# Reactivity of complex communities can be more important than stability

Yuguang Yang[1], Katharine Z. Coyte [2], Kevin R. Foster [3,4] ✉ & Aming Li [1,5] ✉

Understanding stability—whether a community will eventually return to its original state after a perturbation—is a major focus in the study of various complex systems, particularly complex ecosystems. Here, we challenge this focus, showing that short-term dynamics can be a better predictor of outcomes for complex ecosystems. Using random matrix theory, we study how complex ecosystems behave immediately after small perturbations. Our analyses show that many communities are expected to be 'reactive', whereby some perturbations will be amplified initially and generate a response that is directly opposite to that predicted by typical stability measures. In particular, we find reactivity is prevalent for complex communities of mixed interactions and for structured communities, which are both expected to be common in nature. Finally, we show that reactivity can be a better predictor of extinction risk than stability, particularly when communities face frequent perturbations, as is increasingly common. Our results suggest that, alongside stability, reactivity is a fundamental measure for assessing ecosystem health.

A central topic in complex ecosystems is the study of community stability[1-19], which is typically defined as the ability of a system to return to its previous equilibrium after perturbations. This characteristic is considered central for the longevity of ecosystems, and for the persistence of species facing the threat of extinction. Beginning with the seminal work of May[1], a large body of theoretical work has been developed that seeks to understand the causes and consequences of community stability[3,5-19]. Stability is a general and important property of many real-world systems and is not limited to ecology. In evolutionary biology[20-22], for example, stability analysis can be used to predict the outcomes of natural selection in diverse settings, including social networks[21,22]. In engineering[23,24], stability is a key concept in the control theory of many systems, including power grids[24]. Starting in physics and bifurcation analysis, the concept of tipping points has proved useful in multiple disciplines where a loss of stability can be indicative of regime shifts and phase transitions in real-world systems[25-28]. As such, the study of stability can play an important role in predicting the occurrence of critical transitions, which is a

priority in a range of contexts from financial markets to climate change[25-28].

However, an important limitation of most stability analyses is that they focus overwhelmingly on the long-term response of a system to external perturbations[29-31]. That is, stability informs on whether a system will return, but not how this return will happen in practice. This distinction can be critical because the short-term behaviour of a system after perturbation can be opposite to its long-term behaviour, especially when the system exhibits non-normality[29-39].

To better understand such short-term behaviour, the concept of reactivity was introduced[30,31,33-38]. Reactive systems are those that can initially amplify small perturbations, which can lead to large fluctuations in population sizes (Fig. 1). Such large-scale fluctuations in abundances can put species at risk, even though a system is formally stable. It is important, therefore, that we understand both reactivity and stability if we are to predict the responses of ecosystems to perturbations. The importance of reactivity has been recognised in epidemiology[40,41], food security[42] and other networked systems[38,39], where short-term responses can also be critical to outcomes. In

[1]Center for Systems and Control, College of Engineering, Peking University, 100871 Beijing, China. [2]Division of Evolution and Genomic Sciences, Faculty of Biology, Medicine and Health, University of Manchester, Manchester M13 9PT, UK. [3]Department of Biology, University of Oxford, Oxford OX1 3SZ, UK. [4]Department of Biochemistry, University of Oxford, Oxford OX1 3QU, UK. [5]Center for Multi-Agent Research, Institute for Artificial Intelligence, Peking University, 100871 Beijing, China. ✉e-mail: kevin.foster@biology.ox.ac.uk; amingli@pku.edu.cn

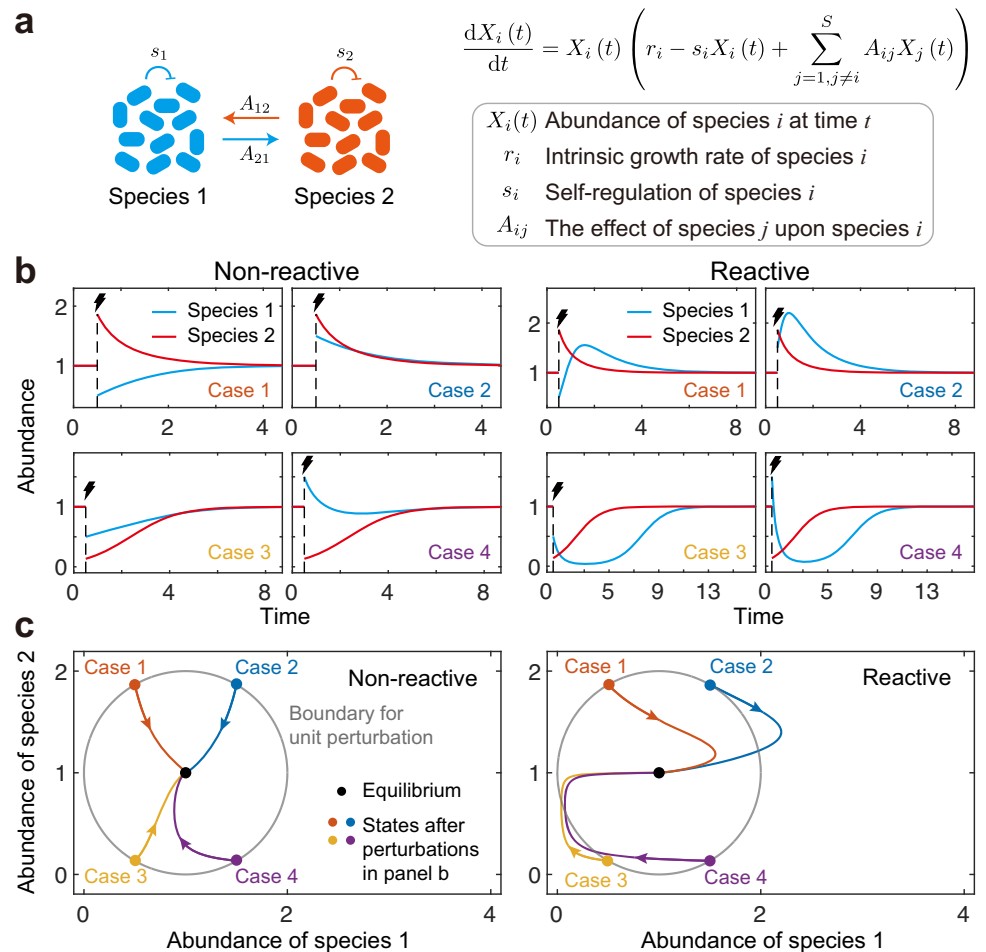

**Fig. 1 | Reactivity captures immediate responses. a** Illustration of a two-species generalised Lotka-Volterra (gLV) community. In this figure, we focus on a case in which species 2 interacts with species 1 unidirectionally (i.e., $A_{12} \neq 0$ and $A_{21} = 0$). **b** Changes of species abundance over time after four different external perturbations for the two-species system. Blue lines are the responses of species 1, and red lines are the responses of species 2. When the interaction strength is relatively low ($s_1 = s_2 = 1$, $A_{12} = 0.3$, left panel) or enlarged ($A_{12} = 3$, right panel), the system can recover from these perturbations, suggesting the system is stable in both cases. **c** The corresponding responses in the phase plane of species abundance. Black dots represent the equilibrium state, and dots of different colours represent immediate states after different perturbations shown in (**b**). Grey circles represent the distance between the equilibrium state and perturbed states in the phase plane of species abundance, suggesting that all perturbed states share the same distance. Trajectories with different colours are responses after different perturbations. When the interaction strength is relatively low (left panel), all perturbations decay initially and all trajectories remain in the grey circle. When the interaction strength is enlarged (right panel), some perturbations increase initially (such as the blue trajectory). Given that the system is stable in both cases, stability cannot describe the instantaneous responses after perturbations. This situation is where the concept of system reactivity is important. If all perturbations decay initially, the system is non-reactive (left panel). If some perturbations are amplified initially, the system is reactive (right panel).

theoretical ecology, the majority of work on reactivity to date has focused on predator-prey systems and small communities containing a few species[34,43,44]. The potential for reactivity in large communities has been demonstrated by Tang and Allesina[31], who found that complex communities can also become reactive before losing stability. However, complex communities come in many forms, with different interaction types and distributions and, across this diversity, we currently lack theory to predict if and when reactivity will be important.

Based on recent progress in random matrix theory[5,7,9,12,14,45–47], here we develop theory to predict when reactivity is expected in complex communities and how this relates to their stability. Our theory allows us to vary the types and strengths of complex interactions in the communities and ask, across a vast diversity of community types, where reactivity is expected. We then apply our modelling to structured food webs and show that reactivity is expected, and demonstrate how reactivity can be a better predictor of species extinctions than typical stability metrics under frequent perturbations.

## Results

### Modelling framework

Typically, an ecosystem composed of $S$ interacting species can be modelled as a continuous-time dynamical system[2,4]

$$\frac{d\mathbf{X}(t)}{dt} = \text{diag}(\mathbf{X}(t))\mathbf{f}(\mathbf{X}(t)), \tag{1}$$

where $\mathbf{X}(t) = (X_1(t), X_2(t), \cdots, X_S(t))^T \in \mathbb{R}^S$ is an $S$-dimensional vector with $X_i(t)$ representing the abundance of species $i$ at time $t$. $\mathbf{f}(\mathbf{X}(t))$ encodes the underlying ecological network and interactions among species. For a specific form of $\mathbf{f}(\mathbf{X}(t))$, namely the generalised Lotka-Volterra model, see Fig. 1a. The dynamical behaviour around a feasible equilibrium $\mathbf{X}^*$ (i.e., all components of $\mathbf{X}^*$ are positive) can be captured by the following linearised equation:

$$\frac{d\mathbf{x}(t)}{dt} = \mathbf{M}\mathbf{x}(t), \tag{2}$$

where $\mathbf{x}(t) = \mathbf{X}(t) - \mathbf{X}^*$ is the deviation from the feasible equilibrium, and $\mathbf{M}$ is the so-called 'community matrix'[1,5] whose element $M_{ij}$ captures the effect that species $j$ has on species $i$ near the equilibrium.

Defined as the maximum instantaneous amplification rate of perturbations[30], reactivity $\mathcal{R}$ is given as follows:

$$\mathcal{R} \equiv \max_{\|\mathbf{x}_0\|_2 \neq 0} \left( \left( \frac{1}{\|\mathbf{x}(t)\|_2} \frac{d\|\mathbf{x}(t)\|_2}{dt} \right) \Big|_{t=0} \right). \qquad (3)$$

Here $\mathbf{x}_0$ is the external perturbation imposed to the system, and $\|\cdot\|_2$ is the 2-norm operator. Mathematically, $\mathcal{R}$ is quantitatively calculated as $\max(\lambda_\mathbf{H})$, where $\lambda_\mathbf{H}$ is the eigenvalue of matrix $\mathbf{H} = (\mathbf{M} + \mathbf{M}^T)/2$. If $\mathcal{R} > 0$, the system is reactive, indicating that some perturbations can be amplified initially. If $\mathcal{R} < 0$, the system is non-reactive, suggesting that all perturbations decay initially. Thus, the core of reactivity analysis is to identify $\max(\lambda_\mathbf{H})$. It is worth noticing that since $\mathbf{H}$ is a real symmetric matrix, all eigenvalues are real, meaning that all eigenvalues locate on the real axis of the complex plane.

### Reactivity criteria for large complex ecosystems
Our model allows us to calculate the reactivity criteria for different community types. In particular, species can interact in a range of ways in a given community—including mutualistic (+/+), exploitative (+/−) and competitive (−/−) interactions—and we can ask how these interactions influence reactivity by identifying the corresponding eigenvalue distribution of $\mathbf{H}$. Following the canonical framework in studying system stability[1–12,14,15,17–19], our analyses are focussed on the properties of the community matrix, which allows us to provide general results that do not rest on particular assumptions of the population dynamics of species. However, we also provide examples where we illustrate dynamics based upon the standard Lotka-Volterra model (e.g., Fig. 1a).

Random interaction distribution: The typical community type studied in theoretical ecology is a 'random' community[1,5]—that is, one where two species $i$ and $j$ interact with probability $C$, and the interaction strengths $M_{ij}$ and $M_{ji}$ take the value of a random variable $Z$ with mean 0 and variance $\sigma^2$ respectively and independently. The diagonal terms, representing self-regulation, are all set to $-d$. For large random systems, the eigenvalues of $\mathbf{H}$ are contained in a line segment with length $2\sigma\sqrt{2SC}$ centred at $(-d, 0)$ (Fig. S1, see Supplementary Note 1 for detailed derivation). This brings $\mathcal{R} = \max(\lambda_\mathbf{H}) = -d + \sigma\sqrt{2SC}$. Reactivity requires that $\mathcal{R} > 0$, which leads to the reactivity criterion: $\sigma\sqrt{2SC} > d$ (Fig. 2a).

Exploitative interactions: In a community with exploitative interactions (e.g., predator-prey, or host-parasite), two species $i$ and $j$ also interact with probability $C$, but interaction strengths have opposite signs: one interaction strength takes the value of $|Z|$ while the other is sampled from $-|Z|$. For large exploitative communities, the eigenvalues of $\mathbf{H}$ are distributed in a line segment with length

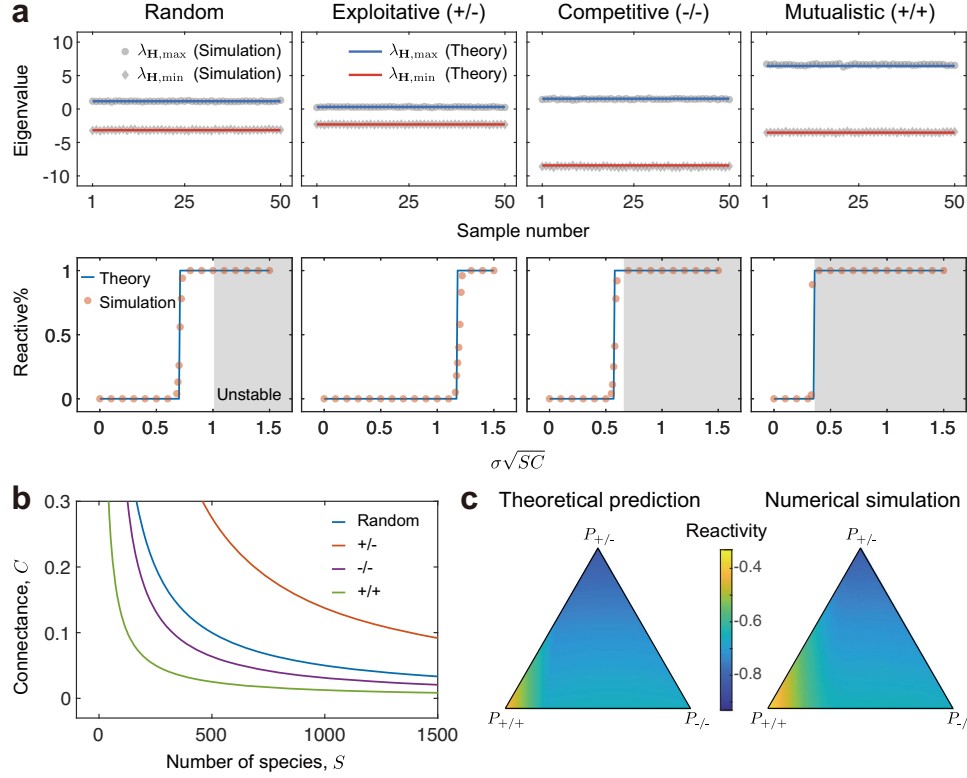

**Fig. 2 | Reactivity criteria for large complex ecosystems. a** Predicting the eigenvalue distribution of $\mathbf{H}$ and corresponding reactivity profiles. In the first row, the maximum (grey circles) and minimum eigenvalues (grey diamonds) of 50 randomly generated communities are plotted. Blue and red lines are theoretical predictions of maximum and minimum eigenvalues, respectively. In the second row, we systematically vary $C$ to obtain $\sigma\sqrt{SC}$ spanning [0,1.5]. Orange dots represent the percentage of reactive communities out of 100 samples from numerical simulations. Blue lines are the corresponding theoretical predictions for the numerical percentage. Grey regions show unstable communities. In all cases, phase transitions from non-reactivity to reactivity are well predicted by our theory. **b** Reactivity criteria for different types of communities. Curves with different colours are critical $C$–$S$ curves for different communities, and combinations of $S$ and $C$ below each curve lead to non-reactive communities. Different types of communities form a strict hierarchy from exploitative communities (most likely to be non-reactive) to mutualistic communities (most likely to be reactive). **c** Extension of our theory to communities with mixed interaction types. Left part of this panel gives theoretical predictions and right part of this panel shows results from numerical simulations. Each data point in the right part is an average of 50 randomly generated communities with the same set of parameters. In this figure, we have $d = 1$. In the first row of (**a**), we have $S = 150$, $C = 0.25$, $\sigma = 0.25$. In the second row of (**a**), we have $S = 250$, $\sigma = 0.1$. In (**b**), we have $\sigma = 0.1$. In (**c**), we have $S = 150$, $C = 0.1$, $\sigma = 0.05$.

$2\sqrt{2SC(\sigma^2 - \mathbb{E}^2(|Z|))}$. The centre of this line segment is $(-d, 0)$ (Fig. S1, see Supplementary Note 1 for detailed derivation). Thus, we have $\mathcal{R} = \max(\lambda_{\mathbf{H}}) = -d + \sqrt{2SC(\sigma^2 - \mathbb{E}^2(|Z|))}$ and the reactivity criterion becomes: $\sqrt{2SC(\sigma^2 - \mathbb{E}^2(|Z|))} > d$ (Fig. 2a).

Competition and mutualism: In a competitive (or mutualistic) community, pairwise interaction strengths have the same sign: negative for competition and positive for mutualism. For competitive communities, interaction strengths are sampled from $-|Z|$, while for mutualistic communities, interaction strengths are sampled from $|Z|$. In both cases, all $\lambda_{\mathbf{H}}$ except an outlier (which is approximately equal to the row sum of $\mathbf{H}$) are contained in a line segment when the community size is large (Fig. S1, see Supplementary Note 1 for detailed derivation). For large competitive communities, this outlier is on the left side of the line segment and reactivity is thus determined by the right endpoint of the line segment. We then have $\mathcal{R} = -d + C\mathbb{E}(|Z|) + \sqrt{2\sigma^2 SC + 2\mathbb{E}^2(|Z|)SC(1 - 2C)}$, and the corresponding reactivity criterion is $C\mathbb{E}(|Z|) + \sqrt{2\sigma^2 SC + 2\mathbb{E}^2(|Z|)SC(1 - 2C)} > d$. For large mutualistic communities, this outlier is on the right side of the line segment and thus determines reactivity: $\mathcal{R} = -d + (S - 1)C\mathbb{E}(|Z|)$. The reactivity criterion now becomes: $(S - 1)C\mathbb{E}(|Z|) > d$ (Fig. 2a).

These reactivity criteria allow us to compare how interaction types influence the reactivity of a community. This analysis reveals that, as for stability, reactivity is strongly dependent on the nature of the interactions within a community. Focussing on reactivity alone, we observe a strict hierarchy from mutualistic communities, which are the most likely to be reactive, through to exploitative communities, which are the least likely (Fig. 2b). Further extension of our theory to communities in which different interactions are mixed with arbitrary proportions (see Methods and Supplementary Note 1) also validates this finding (Fig. 2c, Fig. S2). This result follows the intuition that exploitative interactions, such as predator-prey interactions, introduce negative feedbacks between pairwise species, which will limit the potential for amplification of a perturbation. By contrast, competitive and mutualistic interactions introduce positive feedbacks, which will tend to amplify perturbations. However, note that these positive feedbacks also tend to destabilise communities. As we discuss further below, this effect means that reactive competitive communities and reactive mutualistic communities also tend to be unstable.

## The impact of weak interactions on reactivity

In addition to interaction type, interaction strength is considered central to the behaviour of communities and their stability[1,2,4,5,7,9,48,49]. In particular, previous studies suggest that weak interactions are common in natural ecological networks, where they have been hypothesised to be important for ecosystem persistence because they can help to promote ecological stability in some contexts[5,48]. Next we seek to study the influence of weak interactions on reactivity. Using the criteria derived above, we can systematically assess the influence of weak interactions on reactivity. If weak interactions are common in a community[5], we would have a relatively small $\mathbb{E}(|Z|)$. In contrast, if weak interactions are rare[5], we have a relatively large $\mathbb{E}(|Z|)$. As with linear stability[5], this analysis predicts that weak interactions will make competitive communities and mutualistic communities less reactive, have no influence on random communities, and make exploitative communities more reactive. Importantly, we can confirm these key predictions both theoretically and numerically by generating example communities with interspecies interaction strengths drawn from a Gamma distribution, which allows us to control the prevalence of weak interactions, and explicitly calculating their reactivity (see

Supplementary Note 2). Moreover, there is again an intuition to this result: weakening exploitative interactions drives higher reactivity because these interactions are negative feedbacks that tend to help protect against amplification of perturbations. By contrast, weakening mutualistic or competitive interactions lowers reactivity, because these interaction types are reactive.

## The importance of stable but reactive communities

Our analyses have so far focused on what promotes reactivity in complex communities. However, the reactivity is not equally important for all communities. If a community is unstable, then reactivity is of less interest, for the simple reason that the community is unlikely to exist at all. In contrast, determining the reactivity of a stable community may be vital, as doing so can reveal communities that are at first sight robust, but in fact may be very vulnerable to future perturbations. A key goal, therefore, is to identify when stable reactive communities will occur.

The fact that reactivity and stability are determined by the same set of parameters allows us to directly relate these two key properties of ecological communities. We find that as diversity (i.e., number of species, $S$) and complexity (i.e., connectance, $C$) increase, non-reactive stable communities first become reactive stable and then reactive unstable, indicating that the reactive stable state is an intermediate state between non-reactivity and instability (Fig. 3a). This finding is consistent with previous work[31,50], but we also find that this intermediate state is not equally likely for all community types. This result can be seen by plotting both stability criteria and reactivity criteria together for different community types (Fig. 3a). While competitive and mutualistic interactions both promote reactivity, they also promote instability in similar measure with the result that there is limited scope for stable reactive communities. In these cases, therefore, reactivity is not expected to play an important role in the dynamics of stable communities. By contrast, exploitative communities and random communities have larger regions of parameter space where communities are both stable and reactive. Here, reactivity has the potential to play a critical role in community dynamics.

We can quantify the importance of reactivity in stable communities by calculating the area ratio of the reactive stable region (Fig. 3a, red region) to the stable region (Fig. 3a, blue region and red region). This area ratio can be interpreted as the normalised distance between the transition to reactivity and the transition to instability (see Methods and Supplementary Note 3). A low area ratio, as seen for competitive and mutualistic communities, means that these two transitions are close. For such communities, the observation of a reactive state suggests that the corresponding community is close to an unstable state, or is even transitioning into an unstable state. From an ecosystem management perspective, therefore, reactivity would be an early warning of the potential loss of stability. By contrast, the high area ratio seen for random and exploitative communities means that the distance between reactivity and instability is relatively large. Here, reactive states can be far away from unstable states, such that there is less threat to ecosystem stability. However, as we show below, even for these relatively stable communities, there is a threat of extinction when perturbations are frequent.

We also calculate this area ratio for communities of any proportion of the different interaction types (Fig. 3c). Consistent with the findings for the four community types, increasing the proportion of mutualistic interactions or competitive interactions reduces the scope for stable reactive systems (Fig. 3c). By contrast, increasing the proportion of exploitative interactions increases the propensity for stable communities that are also reactive. Moreover, for communities with mixed interaction types, which are expected to be common in nature, there is ample potential for a reactive stable state.

In summary, our theory predicts that reactivity is most important for the dynamics of communities with mixed and exploitative

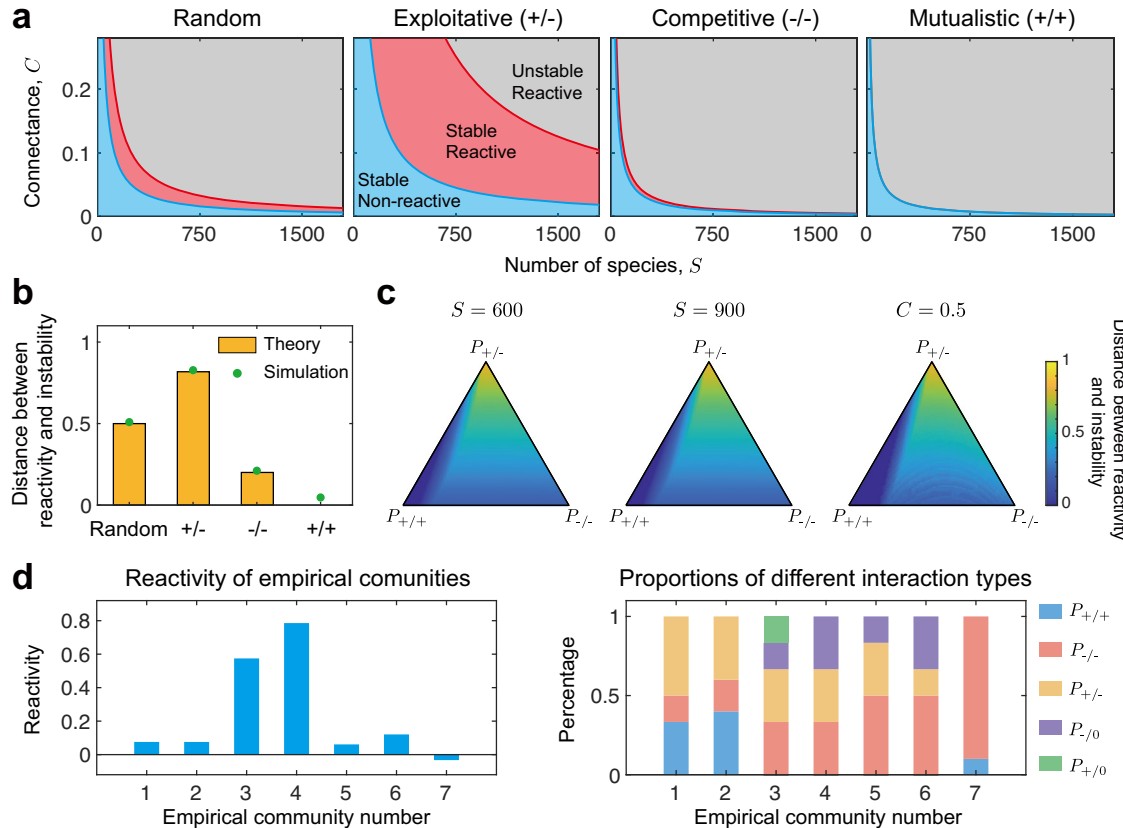

**Fig. 3 | Measuring the distance between reactivity and instability. a** Comparison of the critical $C$–$S$ curves of reactivity and stability for communities with different interaction types. The introduction of reactivity criteria (blue lines) and stability criteria (red lines) can divide the $C$–$S$ plane into 3 parts, where the blue region leads to stable and non-reactive communities, the red region leads to stable and reactive communities, and the grey region leads to unstable and reactive communities. **b** A general view of the distance between reactivity and instability. Here we define the area ratio of the reactive stable region (red region in (**a**)) to the whole stable region (red and blue regions in (**a**)) as the normalised distance between reactivity and instability. Yellow bars are theoretical results, while green dots are numerical results. Each green dot is obtained by measuring the corresponding area ratio in a simulated phase transition diagram (i.e., simulated version of (**a**)). **c** Distance between reactivity and instability for communities with mixed types of interaction (i.e., exploitative interactions, mutualistic interactions, and competitive interactions). When the connectance (or community size) is fixed, the distance is measured by the critical ratio of the number of species (or the ratio of connectance) based on that in (**a**). One can calculate the distance metric, therefore, by either varying community size or connectance to move from the stable non-reactive region, blue line in (**a**), to the unstable reactive region, red line in (**a**). Here we plot this distance for three cases, two of these have fixed community size (left panel and middle panel) while the other one has fixed connectance (right panel). As the proportion of exploitative interactions ($P_{+/-}$) increases, the distance increases. As the proportion of mutualistic interactions ($P_{+/+}$) or competitive interactions ($P_{-/-}$) increases, the distance decreases. And these are consistent with the results given in (**b**). **d** Reactivity analysis of empirical microbial communities. Left part shows the reactivity calculated directly from experimental data, and right part shows proportions of different types of interactions in these communities. Here communities 1 and 2 are mouse microbial communities[51,52], communities 3 to 6 are soil microbial communities[53], and community 7 is an artificially assembled community[49]. In (**a**)–(**c**), we set $d = 1, \sigma = 0.2$.

interactions, where there are large regions of parameter space where communities are both reactive and stable. It is these types of communities, therefore, where one is most likely to see a stable community that, after perturbation, behaves in a manner not predicted by typical stability measures. This said, if one does observe reactivity in a mutualistic or competitive network of species, this may be an early warning sign that the set of species in question are on the brink of a transition to an unstable state.

Our work suggests that reactivity is common in nature. To evaluate this key prediction, we sought data from microbial communities that allow reactivity to be estimated. We identified seven communities, two found within the mammalian gut[51,52], four isolated from the soil[53], and one arbitrarily and artificially assembled in vitro using a mixture of gut and soil-associated microbes[49] (see Supplementary Note 4). We note that these communities are of low diversity (4 or 5 taxa) as compared to those studied with our theory, which is designed for diverse communities. Nevertheless, in line our key prediction that reactivity is important, we find evidence of reactivity in all of the natural communities (Fig. 3d). Moreover, these natural communities also

display a mixture of interaction types, which is associated with stable reactive communities in the theory (Fig. 3d). By contrast, the one artificial community that contains only mutualistic and competitive interactions is non-reactive, which is again consistent with our prediction that these interaction types are less likely to lead to stable reactive systems (Fig. 3d).

## Reactivity in structured food webs
So far we have considered the reactivity of communities with a random interaction network structure (i.e., unstructured interaction network). However, interaction networks of real ecological communities often contain structures, including trophic levels and the potential for groupings of species within a community that interact more strongly than average[5–8,11,12,43,54–58]. We, therefore, want to explore how such structuring influences the potential for reactivity.

To study the impacts of network structure, we implement two widely-used community structure models in our framework: the cascade food web[5,8,12,43,57,58] and the niche food web[5,8,56,57] (see Methods). In the cascade model, species are assumed to form a strict hierarchy, and

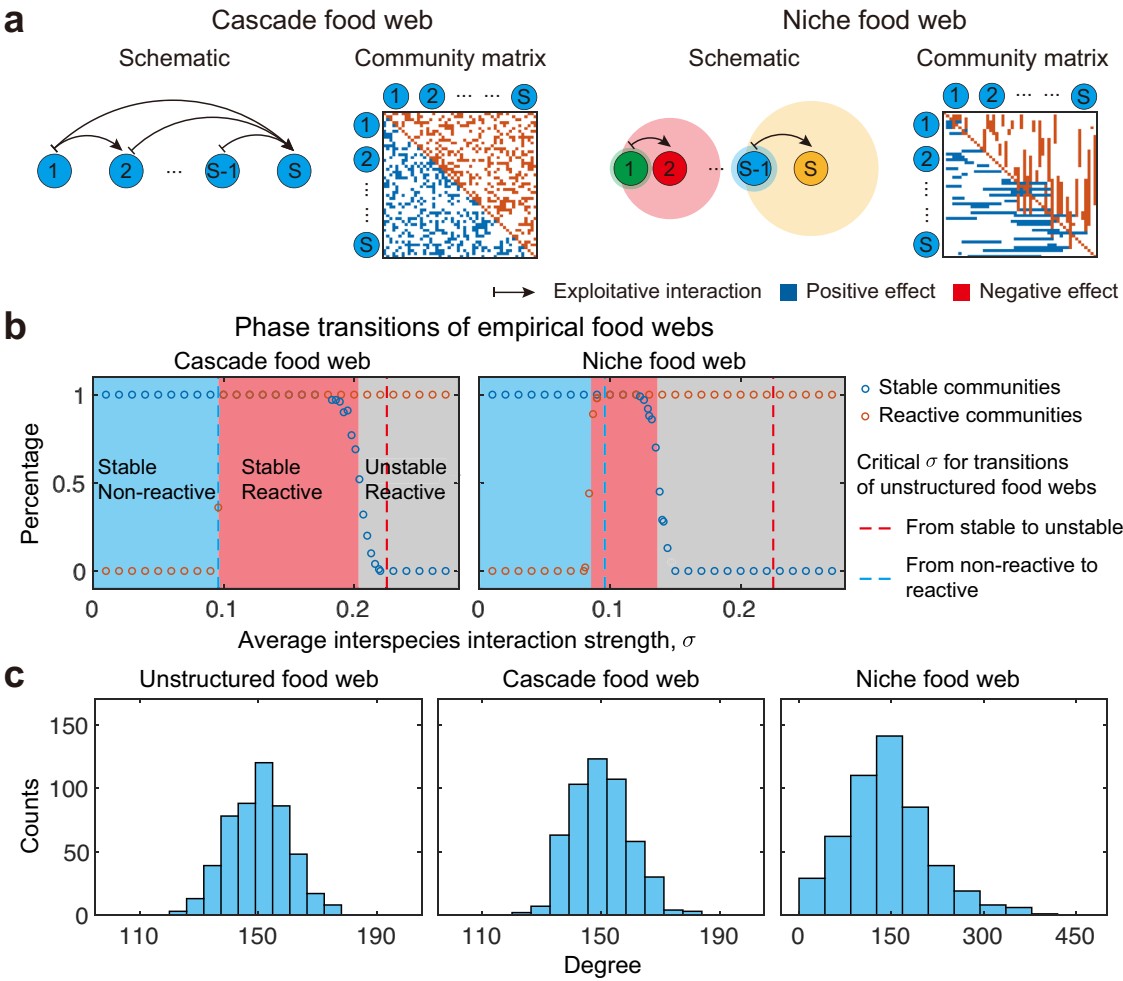

**Fig. 4 | Structured food webs display reactivity. a** Schematic illustration of cascade and niche food webs. In the cascade model, trophic levels are introduced, i.e., species with higher ranks predate species with lower ranks with a fixed probability. In the niche model, species predate a subset of species within a defined range or niche (shaded circles with different colours). The corresponding community matrices are presented with positive (blue) and negative (red) interactions. The construction methods for these two food web models are provided in Methods. **b** Phase transitions of cascade and niche food webs. The blue region marks stable and non-reactive communities, red region marks stable and reactive communities, and grey region marks unstable and reactive communities. Red (blue) hollow dots are the percentage of reactive (stable) communities from numerical simulations, obtained from 50 randomly constructed communities for each point in parameter space. For comparison, the vertical red (blue) dashed lines show the critical average interspecies interaction strength (i.e., $\sigma$) for the transition to instability (reactivity) of the corresponding unstructured food webs. As $\sigma$ increases, these two food webs also experience two phase transitions: one is from stable non-reactive state to stable reactive state, the other is from stable reactive state to unstable state. The cascade model shares the same critical $\sigma$ with the unstructured case to move into a stable reactive state, while niche model requires a lower average strength of interspecies interaction to move into a stable reactive state. **c** Degree distribution for these three types of food webs. In (**b**) and (**c**), community parameters are: $S = 500, C = 0.3, d = 1$.

species with higher rank predate species with lower rank with a fixed probability (Fig. 4a). In the niche model, species predate all species within their predation range (Fig. 4a). Importantly, despite the introduction of network structure, we still observe large regions of parameter space where communities are stable and reactive (Fig. 4b). Indeed, the introduction of structure in the niche model even makes communities more reactive, i.e., reactive communities occur for weaker levels of interspecies interaction than in the equivalent unstructured communities (recall that these communities are based on exploitative interactions which tend to limit reactivity, above). Put another way, it takes lower interspecies interaction strengths to render a community reactive with niche structuring than without.

Our work shows that, as compared with an unstructured equivalent, cascade food webs do not influence system reactivity, while a niche food web structure makes the system more reactive (Fig. 4b). We find that this difference between the two food web types is explained by the different structural features introduced by each. The cascade model introduces trophic levels, which only influence the arrangement

of pairwise positive and negative elements in the community matrix **M** (the non-zero negative elements are in the upper-triangular part, while the non-zero positive elements are in the lower-triangular part, Fig. 4a). This single feature is mitigated in $\mathbf{H} = \left(\mathbf{M} + \mathbf{M}^{\mathrm{T}}\right)/2$, and, as a result, the cascade model and its unstructured counterpart yield the same **H**, and therefore the same reactivity. The niche model is different in that it introduces intervality (that is, each predator consumes preys that are adjacent in the hierarchy) and a broader degree distribution (Fig. 4c, degree of a species is the number of its interacting species) alongside introducing trophic levels[8]. These properties not only change the arrangement of pairwise positive and negative elements in the community matrix but also the topological structure of the underlying interaction network. This change of topological structure influences matrix **H**. Thus, $\mathbf{H}_{\mathrm{niche}}$ is different from $\mathbf{H}_{\mathrm{unstructured}}$, which leads to a more reactive community. Further analysis of some variants of cascade model (interval cascade model, cascade model with broad degree distribution, and interval cascade model with broad degree distribution, see Supplementary Note 5 and Figs. S5–S7) suggests that the

broader degree distribution is the key structural feature that drives increased reactivity.

## Self-regulation and reactivity of complex ecosystems

A key factor known to influence community stability is the strength of density dependent regulation exhibited by the populations of each species in a community[4,5,12], also known as self-regulation (i.e., $M_{ii}$). Broadly speaking, increased self-regulation is stabilising and, therefore, is predicted to shift communities to a more stable state[5,9]. As might be expected, therefore, it is also the case that increasing self-regulation can lead to a less reactive community (since it is clear from the theoretical expressions that more negative $M_{ii}$, i.e., larger self-regulation strength, can bring a smaller reactivity value). However, our model assumes that all species have identical levels of self-regulation. In practice, the expectation[12,59,60] is that there will be variability in self-regulation between species with the possibility of some species that have very limited self-regulation, which are instead regulated by their interactions with the wider community. We, therefore, sought to understand the potential impacts of such self-regulation heterogeneity on system reactivity, and whether a system can remain non-reactive when some species do not self-regulate. Here again, we were able to leverage developments in random matrix theory[12,45–47] in order to incorporate heterogeneous self-regulation strengths and non-self-regulating species into our theoretical framework of system reactivity (Methods and Supplementary Note 6).

With this method in place, we can vary the heterogeneity of self-regulation in unstructured networks as the standard deviation of self-regulation strength $\sigma_d$, which reveals that increasing self-regulation heterogeneity makes the system more reactive (Fig. 5a and Fig. S10, see Supplementary Note 6 for theoretical analysis). We next ask whether the introduction of network structure alters this prediction, using the case of a cascade food web. Specifically, here we consider three cases. In the first, the strengths of self-regulation have no relation to trophic levels (disorganised case). The second assumes that species at higher trophic levels possess higher self-regulation strengths (positive case). In the third, species at higher trophic levels possess lower self-regulation strengths (negative case). We find that all these three cases yield the same results as unstructured food webs (Fig. 5b). Our findings on the importance of heterogeneity in self-regulation for reactivity, therefore, are robust to these changes in network structure.

Similarly, we can explore the influence of non-self-regulating species on system reactivity in communities of otherwise homogeneous self-regulation strengths. Strikingly, we find that even a single species without self-regulation can make a non-reactive system reactive, regardless of the self-regulation strength of other community members (Fig. 5c, Fig. S10, see Supplementary Note 6 for theoretical analysis). For most community types, this removal of self-regulation typically also renders communities unstable. However, interestingly, we find that exploitative systems are able to remain stable even when multiple species no longer self-regulate, yet become increasingly reactive as the number of non-self-regulating species increases. This finding holds in the presence of trophic levels (Fig. S10), and when those species that do self-regulate do so in a heterogeneous manner (Fig. S10). Together, these analyses suggest that all species must self-

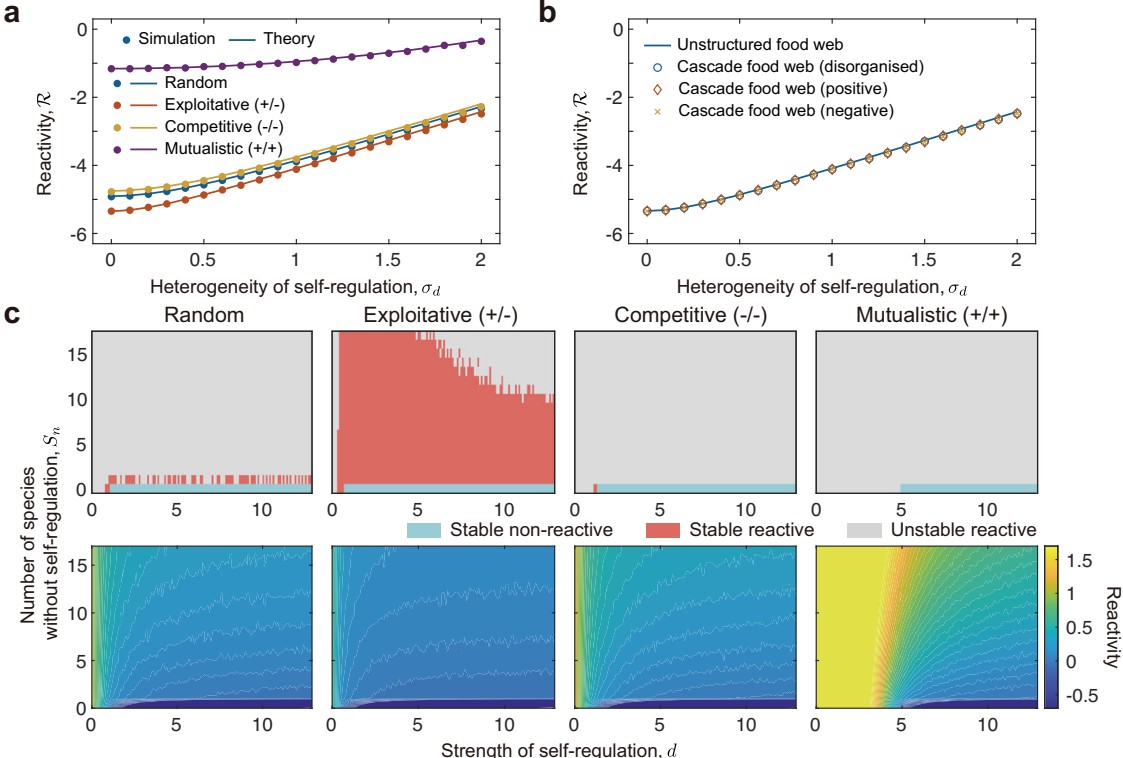

**Fig. 5 | Influence of self-regulation on system reactivity. a, b** Influence of self-regulation heterogeneity on system reactivity. Self-regulation heterogeneity is quantified by the standard deviation of self-regulation strengths. Solid lines are theoretical predictions, symbols (dots, circles, diamonds, and crosses) are results from numerical simulations. 'Disorganised' refers to the case where self-regulation strengths have no relation to trophic levels, 'positive' refers to the case where species at higher trophic levels have higher self-regulation strengths, and 'negative' refers to the case where species at higher trophic levels have lower self-regulation strengths. Note that in these two panels, all communities concerned are stable.

**c** Influence of non-self-regulating species on system reactivity. Upper panels show average community state, with the blue region indicating communities are on average stable and non-reactive, the red region indicating stable and reactive, and the grey region indicating unstable and reactive. Lower panel shows the average reactivity values for each parameter combination. In (**a**) and (**b**), the average self-regulation strength is $d_{mean} = 6$, and self-regulation strengths are sampled from a uniform distribution. Other community parameters in this figure are $S = 200, C = 0.3, \sigma = 0.1$. Each simulation data point is the average of 50 randomly constructed communities for the specific parameter combination.

regulate with relatively high strengths in order to prevent community reactivity.

## Reactivity can be a better predictor of species extinctions than stability under frequent perturbations

Finally, to investigate the potential importance of reactivity, we study the impacts of different perturbation regimes on the structured and unstructured networks (Fig. 6 and Figs. S11–S14). For single perturbations, as expected, the effects of reactivity are that the return time after perturbation is increased relative to networks that are non-reactive. This increase in return time is seen across the three types of networks—unstructured, cascade and niche—and the result is that the systems return to their original abundances much more slowly than in the non-reactive case (Fig. 6a, top). Nevertheless, they do all return in line with the notion of a reactive but stable state. The introduction of frequent perturbations, however, results in a qualitative shift in predictions. Now, the effects of reactivity dominate the dynamics that are observed

and, importantly, reactive communities are more likely to experience species extinctions than non-reactive ones (Fig. 6a, bottom and Fig. 6b). Modelling other types of frequent perturbations further supports this claim (Figs. S11–S14). Under such conditions, therefore, reactivity becomes more important than stability in predicting the persistence of species over time.

Frequent perturbations can also be modelled through the theoretical framework for modelling ecological variability of Arnoldi et al.[61]. This modelling framework is a linearised model and is thus valid for systems operating near equilibrium. As a result, it is not suited for studying extinctions directly, as these tend to occur when a system is far from equilibrium. Nevertheless, one can use variability in abundances over time to capture transient responses to perturbations, where the magnitude reflects the tendency of a community to change in time. In this way, variability can be a good indicator of the risk that an ecosystem will experience species loss and system collapse[35,61]. Inspired by this approach[61], we constructed a perturbed community

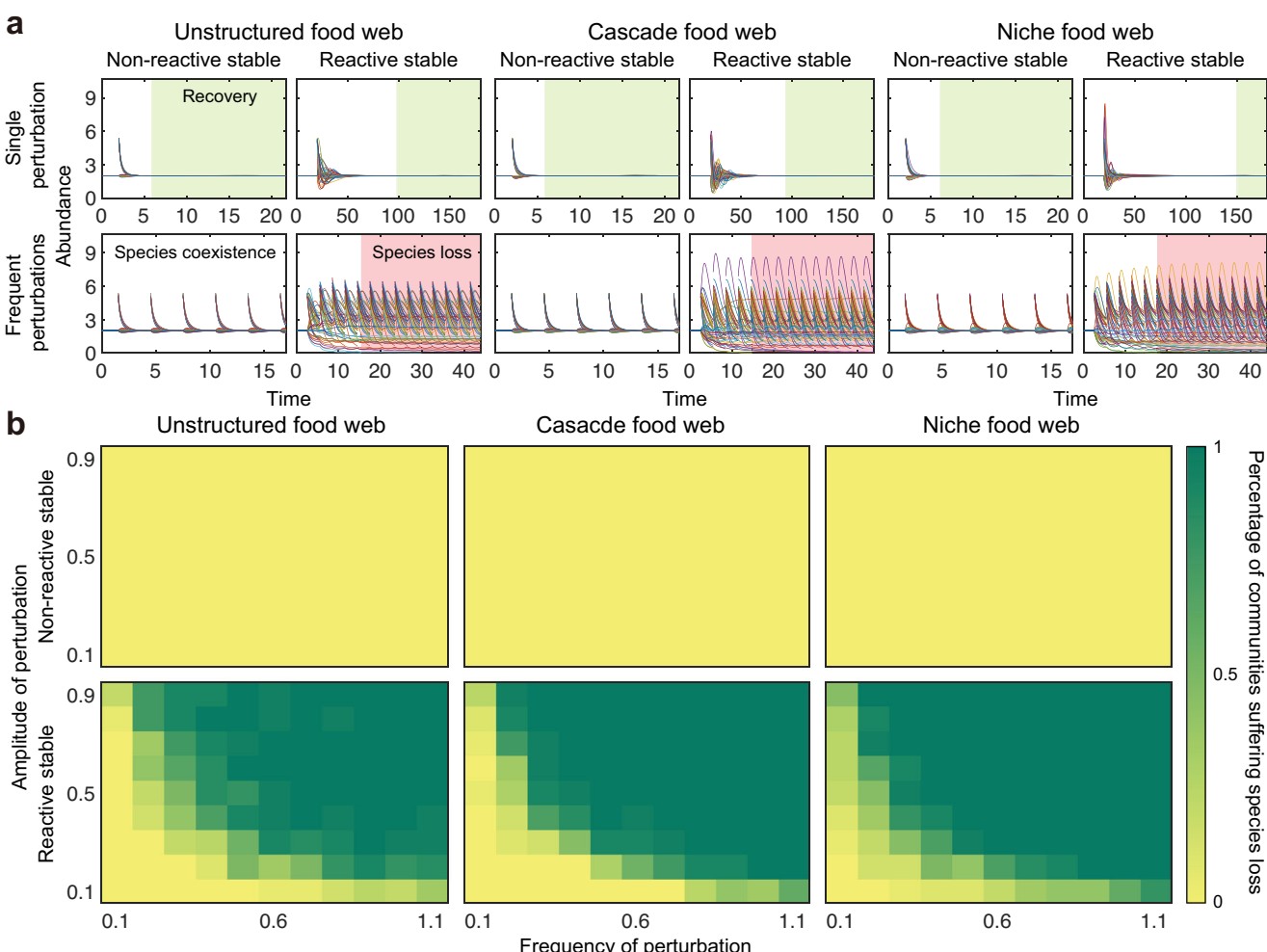

**Fig. 6 | Reactivity predicts species persistence better than stability under frequent perturbations. a** Responses of different food webs to perturbations. Each line in (**a**) captures the change in abundance of each species after the perturbations. Green regions indicate that the community recovers to its equilibrium, and red regions suggest that the community suffers species loss. Although both non-reactive stable and reactive stable communities can recover from a single perturbation (upper row of (**a**)), reactive stable communities can experience species loss following frequent perturbations (bottom row of (**a**)). **b** Overall view of species loss in typical food webs under frequent perturbations with different strengths and frequencies. Colours represent the percentage of communities suffering species loss from numerical simulations. To obtain the corresponding percentage, we

randomly construct 20 communities and count how many communities can persist under frequent perturbations. In each analysis, community dynamics are simulated using the gLV model, with a species regarding as extinct if its abundance falls below 1% of its equilibrium abundance, and community parameters are: $S = 50$, $C = 0.2$, $\sigma = 0.05$. Per capita self-regulation strength $s = 1$ for non-reactive stable communities, and $s = 0.1$ for reactive stable communities. Frequent perturbations are imposed by increasing the abundances of 30 randomly picked species with a fixed frequency but the same pattern is seen for other perturbation types, see Figs. S11–S14. For communities in (**a**), equilibrium abundance is 2 for each species, in (**b**), equilibrium abundance is 1 for each species and simulation time is 500 units.

model and explored the relationship between variability and reactivity (Supplementary Note 7). This reveals that, as compared with non-reactive communities, the variability of a stable reactive community is high, which is consistent with our finding that stable reactive communities are more vulnerable to species loss and system collapse under frequent perturbations (Figs. S15–S18).

## Discussion

The study of community stability is central to ecology. Stability is considered to be so important because most natural communities will face frequent perturbations, and it is the stable communities that are expected to maintain species diversity in the face of these perturbations. However, this view neglects the potential for reactivity in stable communities, which can amplify perturbations and dominate the short-term responses to perturbations. Here, we have explored the conditions that lead to reactivity in diverse communities as a function of the sign and strength of the complex interactions among their constituent species. By deriving analytical reactivity criteria, we show that interaction types can be critical for community reactivity. Mutualistic and competitive communities are typically only reactive when they are unstable and here, the observation of reactivity is likely to indicate the potential for community collapse. However, for exploitative and mixed interaction types, we find large regions where communities are predicted to be both stable and reactive. This prediction is robust to the introduction of community structure, where implementing a cascade or niche model leads again to large regions of reactivity within stable communities.

The link between reactivity and short-term dynamics can be lessened in some contexts. The initial amplification rate strongly depends on perturbation direction, and thus being reactive does not imply that all perturbations are amplified initially[37]. Another context in which reactivity can be decoupled from short-term dynamics is when there is a wide distribution of species abundances. Arnoldi et al. studied the influence of rare species and found that rare species tend to influence the measure of system stability even though they do not greatly affect system dynamics in practice[37]. In the same vein, we explored the influence of rare species on our measure of system reactivity within a range of different community types (see Supplementary Note 8 and Fig. S19). This work shows that regardless of community type, rare species can again influence system reactivity values, even though again these species would not in practice have a large impact on system dynamics (Fig. S20). To analyse such cases and gain a comprehensive view on short-term system dynamics after perturbations, one should consider other measures, such as median return rate proposed by Arnoldi et al., along with system reactivity and stability[37]. However, because our main analyses did not include such rare species, the measure of reactivity we have employed here should map well to short-term dynamics.

Rather than focussing on the case of rare (low abundance) species, one can instead ask more generally how differences in species abundances influence community ecology. Recent work by Gibbs et al. extending classic work on system stability revealed that species abundances do not qualitatively affect stability in a Lotka-Volterra framework[14]. That is, as long as an interaction matrix $\mathbf{A}$ is stable, the community matrix $\mathbf{M} = \mathrm{diag}\left(\mathbf{X}^{*}\right)\mathbf{A}$ will also be stable for any feasible equilibrium $\mathbf{X}^{*}$. This result led us to wonder whether similar rules might apply to reactivity. To investigate the potential role of species abundances on reactivity, we conducted additional theoretical analyses and numerical calculations (Supplementary Note 8, Figs. S21–S23). While our theory is only an approximation (Supplementary Note 8), it fits well with our numerics (Figs. S21, S22) and suggests that the reactivity of $\mathbf{A}$ implies the reactivity of corresponding $\mathbf{M}$ but the non-reactivity of $\mathbf{A}$ cannot guarantee the non-reactivity of corresponding $\mathbf{M}$ (Supplementary Note 8, Fig. S23). This result is important in that it implies that accounting for variability in species abundances will only increase the scope for reactivity beyond that which we have predicted here.

We also find that the degree of self-regulation by individual species in a community can be important for system reactivity. Self-regulation has long been known to have the potential to promote system stability, with some studies suggesting that the majority of species need to have strong self-regulation strengths to maintain stability[12]. Our work suggests that to prevent reactivity the requirements for self-regulation are even stricter, with a requirement that all species must self-regulate with relatively high strengths. Given we also find conditions where reactivity drives species extinctions, this finding suggests that self-regulation may play a more important role in ecosystem health than previously appreciated.

Our work shows that communities with mixed interactions and trophic structure have ample potential for reactivity, suggesting that reactive states can exist in many naturally-occurring systems. In support of this conclusion, previous work has highlighted the potential for reactivity through analysis of ecological data, including in planktonic lake communities[62] and insect populations[63]. When perturbations are rare, these dynamics are transient and stability becomes a reasonable predictor of the state of a given community. In other cases, however, the observation of reactivity will be an early warning sign of extinction risk, one that can potentially be assessed more quickly than stability property in ecological data. Indeed, for many ecosystems, the expectation is that perturbations will be frequent, something that may increase with the impacts of climate change and other anthropogenic factors[64–67]. Here, short-term responses can dominate and it is reactivity, rather than stability, that is the key to both ecological dynamics and extinction risk. Our work suggests that reactivity and stability need to be considered side-by-side if we are to understand, and predict, complex systems.

## Methods
### Constructing community matrices for communities with mixed interaction types

For communities with mixed interaction types, two species still interact with probability $C$. With probability $P_{+/+}$, two species interact in mutualistic manner, and the interaction strengths $M_{ij}$ and $M_{ji}$ take the value of $|Z|$ respectively and independently. With probability $P_{-/-}$, two species interact in competitive manner, and the interaction strengths $M_{ij}$ and $M_{ji}$ take the value of $-|Z|$ respectively and independently. With probability $P_{+/-}$, two species interact in exploitative manner, and the interaction strengths $M_{ij}$ and $M_{ji}$ have opposite signs: one takes the value of $|Z|$ while the other takes the value of $-|Z|$. The diagonal terms $M_{ii}$ are all set to $-d$. Note that $P_{+/+} = 1$ leads to a mutualistic community, $P_{-/-} = 1$ leads to a competitive community, and $P_{+/-} = 1$ leads to an exploitative community. The statistics of the community matrix for communities with mixed interaction types can be extracted as

$$\begin{cases} \mathbb{E}\left(M_{ij}\right)_{i \neq j} = C\mathbb{E}(|Z|)\left(P_{+/+} - P_{-/-}\right), \\ \mathrm{Var}\left(M_{ij}\right)_{i \neq j} = C\sigma^2 - \left(\mathbb{E}\left(M_{ij}\right)_{i \neq j}\right)^2, \\ \mathbb{E}\left(M_{ij}M_{ji}\right)_{i \neq j} = C\mathbb{E}^2(|Z|)\left(P_{+/+} + P_{-/-} - P_{+/-}\right). \end{cases} \tag{4}$$

### Reactivity criterion for communities with mixed interaction types

For simplicity, we denote $\mathbb{E}(M_{ij})_{i \neq j} = E$, $\mathrm{Var}\left(M_{ij}\right)_{i \neq j} = V$, and $\mathbb{E}(M_{ij}M_{ji})_{i \neq j} = \rho$. Based on these statistics, we have $\mathbb{E}(H_{ij})_{i \neq j} = E$, $\mathrm{Var}\left(H_{ij}\right)_{i \neq j} = (V + \rho - E^2)/2$, and $\mathbb{E}(H_{ij}H_{ji})_{i \neq j} = (V + \rho + E^2)/2$. According to random matrix theory and low-rank perturbation theorem[5,7,12,45,46], the eigenvalues of $\mathbf{H}$ are contained in a line segment

with length $2\sqrt{2S(V+\rho-E^2)}$ centred at $(-d-E,0)$ when $|E|\leq\sqrt{(V+\rho-E^2)/(2S)}$. In this case, reactivity is

$$\mathcal{R} = -d - E + \sqrt{2S(V+\rho-E^2)}. \tag{5}$$

When $|E| > \sqrt{(V+\rho-E^2)/(2S)}$, all but one eigenvalues are still distributed in this line segment, and the outlier is approximated as $\lambda_{\mathbf{H},\text{outlier}} = -d + (S-1)E$. In this case, reactivity is

$$\mathcal{R} = \max\left(-d - E + \sqrt{2S(V+\rho-E^2)}, -d + (S-1)E\right). \tag{6}$$

See Supplementary Note 1 for detailed derivation.

## Measuring the distance between the transition to reactivity and the transition to instability

As stated in the manuscript, the normalised distance (henceforth, ND) between the transition to reactivity and the transition to instability can be defined as the area ratio of the reactive stable region to stable region. For random communities, the stability criterion is $\sigma\sqrt{SC} < d$, leading to the function describing the critical $S-C$ curve for instability $S = d^2/(C\sigma^2)$. The area of the stable region in $S-C$ plane is obtained as $A_{\text{stable,random}} = (d^2/\sigma^2)\ln(C_2/C_1)$. Similarly, we can derive the function depicting the critical $S-C$ curve for reactivity $S = d^2/(2C\sigma^2)$, and the area of the non-reactive region in $S-C$ plane $A_{\text{non-reactive,random}} = (d^2/(2\sigma^2))\ln(C_2/C_1)$. The normalised distance between the transition to reactivity and the transition to instability (i.e., the area ratio of reactive stable region to stable region) is then $\text{ND}_{\text{random}} = 1 - A_{\text{non-reactive,random}}/A_{\text{stable,random}} = 1/2$ (Fig. 3b). When mutualistic interactions are preponderant, the stability criterion and reactivity criterion are $(S-1)E < d$ and $(S-1)E > d$, respectively. Thus, the critical $S-C$ curves for instability and reactivity are the same, meaning that a reactive mutualistic community is always unstable (Fig. 3b). Otherwise, the stability criterion and reactivity criterion are $-E + \sqrt{SV}(1+(\rho-E^2)/V) < d$ and $-E + \sqrt{2S(V+\rho-E^2)} > d$, respectively. The critical $S-C$ curves are $S = V(d+E)^2/(V+\rho-E^2)^2$ and $S = (d+E)^2/(2(V+\rho-E^2))$ respectively, leading to the distance between the transition to reactivity and the transition to instability $\text{ND}_{\text{mixed}} = 1 - (1 + r(M_{ij},M_{ji})_{i\neq j})/2$, where $r(M_{ij},M_{ji})_{i\neq j} = (\rho-E^2)/V$ (note that here the distance is obtained by fixing system connectance). Clearly, a positive $r(M_{ij},M_{ji})_{i\neq j}$ (i.e., competitive and mutualistic interactions are preponderant) leads to a relatively low distance (lower than 1/2, Fig. 3b, c), and a negative $r(M_{ij},M_{ji})_{i\neq j}$ (i.e., exploitative interactions are preponderant) leads to a relatively large distance (larger than 1/2, Fig. 3b, c). For detailed derivation and analysis, see Supplementary Note 4.

In simulations, critical values for phase transitions (e.g., the boundaries between regions with different colours in Fig. 4b) are determined as follows, taking the transition from a non-reactive state to a reactive state as an example. First, we identify the average interspecies interaction strength value $\sigma_1$, at which all simulated communities are non-reactive, such that when $\sigma > \sigma_1$, not all simulated communities are non-reactive. Next, we identify the average interspecies interactions strength value $\sigma_2$, at which all simulated communities are reactive, such that when $\sigma < \sigma_2$, not all simulated communities are reactive. The critical $\sigma_{\text{r}}$ is then estimated as $\sigma_{\text{r}} = (\sigma_1 + \sigma_2)/2$. Similarly, we can estimate the critical $\sigma$ for transition from a stable state to an unstable state.

## Reactivity analysis of empirical microbial communities

We analyse the reactivity of seven different microbial communities[49,51–53] from previous empirical studies. For these communities, the researchers have already inferred the true interaction networks ($A_{ij}$) and intrinsic growth rates ($r_i$). For a given community, we then perform the following steps: 1) Using the empirically derived $A_{ij}$ and $r_i$ parameters, we identify the underlying equilibrium abundances of each species within the community, then determine the empirical community matrix $\mathbf{M}_{\text{e}}$. 2) The true reactivity can be drawn directly from $\mathbf{M}_{\text{e}}$, via calculating the eigenvalues of $\mathbf{H}_{\text{e}} = (\mathbf{M}_{\text{e}} + \mathbf{M}_{\text{e}}^{\text{T}})/2$ (blue bars, left part of Fig. 3d). For detailed information, please see Supplementary Note 4.

## Constructing community matrices for cascade food web and niche food web

The construction algorithm[5] for a cascade food web is: i) For each entry in the lower-triangular part of $\mathbf{M}$ (i.e., $M_{ij,i>j}$), we draw a random value $p$ from a unifrom distribution $U[0,1]$. ii) If $p \leq C$, we draw $M_{ij,i>j}$ from an half-normal distribution $|N(0,\sigma^2)|$ and $M_{ji,i>j}$ from a negative half-normal distribution $-|N(0,\sigma^2)|$. $C$ is the desired level of connectance. iii) If $p > C$, we set 0 to both $M_{ij,i>j}$ and $M_{ji,i>j}$. iv) Set all diagonal terms to $-d$ (i.e., $M_{ii} = -d$). The construction algorithm[5] for a niche food web is: i) Set a niche value $\eta_i$ for each species and order species according to an increasing niche value order. $\eta_i$ is drawn from a uniform distribution $U[0,1]$. ii) Set a niche radius $r_i = \eta_i\mathcal{B}$ for each species. $\mathcal{B}$ is sampled from a beta distribution $\text{Be}(1,1/C-1)$. $C$ is the desired level of connectivity. iii) Set a niche centre $c_i$ for each species. $c_i$ is sampled from a uniform distribution $U[r_i/2, \min(\eta_i, 1-r_i/2)]$. iv) Construct an adjacency matrix $\mathbf{A}$. If species $i$ is a prey of species $j$ (i.e., $\eta_i \in [c_j - r_j/2, c_j + r_j/2]$), set $A_{ij} = 1$. Otherwise, set $A_{ij} = 0$. v) Construct a sign matrix $\mathbf{P} = -\mathbf{A} + \mathbf{A}^{\text{T}}$. vi) Each element $M_{ij}$ of the community matrix is obtained by multiplying $P_{ij}$ and $Z_{ij}$. $Z_{ij}$ is sampled from an half-normal distribution $|N(0,\sigma^2)|$. vii) Set all diagonal terms to $-d$ (i.e., $M_{ii} = -d$). See Supplementary Note 5 for additional information.

## Constructing community matrices for communities with heterogeneous self-regulation strengths

The differences between the community matrices of communities with homogeneous self-regulation strengths and communities with heterogeneous self-regulation strengths are the diagonal terms (i.e., $M_{ii}$), and the sampling of off-diagonal terms is the same as presented previously. Thus, here we focus solely on the sampling of diagonal terms. When self-regulation strengths and trophic levels are not related, all diagonal terms are sampled from a uniform distribution with mean $d_{\text{mean}}$ and variance $\sigma_d^2$ respectively and independently. Here $d_{\text{mean}}$ is the average self-regulation strength. When self-regulation strengths and trophic levels are related, we first generate $S$ random values $d_1, \cdots, d_S$ from the uniform distribution $U[d_{\text{mean}} - \sqrt{3}\sigma_d, d_{\text{mean}} + \sqrt{3}\sigma_d]$ respectively and independently. When species at higher trophic levels possess higher self-regulation strengths, sort $d_i$ in an ascending order and set $M_{ii} = -d_i$ (note that here $i$ indicates the trophic level). When species at higher trophic levels posses lower self-regulation strengths, sort $d_i$ in a descending order and set $M_{ii} = -d_i$. Note that this construction method can be adapted to cases where self-regulation strengths are sampled from other distributions.

## Constructing community matrices for communities with non-self-regulating species

Here we still focus on the sampling of $M_{ii}$. When self-regulation strengths and trophic levels are not related, we first generate $S_n$ distinct positive integers less than or equal to $S$ randomly: $Q = \{q_1, \cdots, q_{S_n}\}$. For each $i \in Q$, set $M_{ii} = 0$. For each $i \notin Q$, set $M_{ii} = -d$. When species at higher trophic levels self-regulate, we set $M_{ii} = 0$ for $i \leq S_n$ and $M_{ii} = -d$ for $i > S_n$ (note that $i$ indicates the trophic level). When species at higher trophic levels do not self-regulate, we set $M_{ii} = -d$ for $i \leq S - S_n$ and $M_{ii} = 0$ for $i > S_n$.

**Incorporating heterogeneous self-regulation strengths and non-self-regulating species into reactivity analysis**

For convenience, we still denote $\mathbb{E}(M_{ij})_{i \neq j} = E$, $\mathrm{Var}(M_{ij})_{i \neq j} = V$, and $\mathbb{E}(M_{ij}M_{ji})_{i \neq j} = \rho$, which leads to the statistics of matrix $\mathbf{H}$: $\mathbb{E}(H_{ij})_{i \neq j} = E$, $\mathrm{Var}(H_{ij})_{i \neq j} = (V + \rho - E^2)/2$ and $\mathbb{E}(H_{ij}H_{ji})_{i \neq j} = (V + \rho + E^2)/2$.

First, we discuss cases of heterogeneous self-regulation strengths. For the simplicity of theoretical derivation, here we focus on the case where self-regulation strengths are sampled from a uniform distribution $[-d_{\mathrm{mean}} - \sqrt{3}\sigma_d, -d_{\mathrm{mean}} + \sqrt{3}\sigma_d]$. For the classic random community, we have $E = 0$, $V = C\sigma^2$ and $\rho = 0$. We can then identify the rightmost eigenvalue of $\mathbf{H}$ (see Supplementary Note 6 for detailed derivation), which leads to the expression of reactivity

$$\mathcal{R} = \frac{\sqrt{2}}{2}\sigma\sqrt{SC}\left(\frac{d_1 + d_2}{2} + \sqrt{\left(\frac{d_2 - d_1}{2}\right)^2 + 1} + \frac{2}{d_2 - d_1}\tanh^{-1}\left(\frac{d_2 - d_1}{\sqrt{(d_2 - d_1)^2 + 4}}\right)\right),$$

(7)

where

$$\begin{cases} d_1 = \left(-\sqrt{2}d_{\mathrm{mean}} - \sqrt{6}\sigma_d\right)/\sigma\sqrt{SC}, \\ d_2 = \left(-\sqrt{2}d_{\mathrm{mean}} + \sqrt{6}\sigma_d\right)/\sigma\sqrt{SC}. \end{cases}$$

(8)

For communities with mixed interaction types, when $|E| \leq \sqrt{(V + \rho - E^2)/(2S) + \sigma_d^2/S^2}$, we can approximate the rightmost eigenvalue of $\mathbf{H}$ (see Supplementary Note 6 for detailed derivation), which leads to the expression of reactivity

$$\mathcal{R} = \sqrt{\frac{1}{2}S(V + \rho - E^2)}$$
$$\cdot \left(\frac{d_1 + d_2}{2} + \sqrt{\left(\frac{d_2 - d_1}{2}\right)^2 + 1} + \frac{2}{d_2 - d_1}\tanh^{-1}\left(\frac{d_2 - d_1}{\sqrt{(d_2 - d_1)^2 + 4}}\right)\right) - E,$$

(9)

where

$$\begin{cases} d_1 = \left(-d_{\mathrm{mean}} - \sqrt{3}\sigma_d\right)/\sqrt{\frac{1}{2}S(V + \rho - E^2)}, \\ d_2 = \left(-d_{\mathrm{mean}} + \sqrt{3}\sigma_d\right)/\sqrt{\frac{1}{2}S(V + \rho - E^2)}. \end{cases}$$

(10)

When $|E| > \sqrt{(V + \rho - E^2)/(2S) + \sigma_d^2/S^2}$, the expression of reactivity can be approximated as follows (see Supplementary Note 6 for detailed derivation)

$$\mathcal{R} = \max(\mathcal{R}_1, \mathcal{R}_2),$$

(11)

where

$$\mathcal{R}_1 = \sqrt{\frac{1}{2}S(V + \rho - E^2)}$$
$$\cdot \left(\frac{d_1 + d_2}{2} + \sqrt{\left(\frac{d_2 - d_1}{2}\right)^2 + 1} + \frac{2}{d_2 - d_1}\tanh^{-1}\left(\frac{d_2 - d_1}{\sqrt{(d_2 - d_1)^2 + 4}}\right)\right) - E,$$

(12)

and

$$\mathcal{R}_2 = -d_{\mathrm{mean}} + (S - 1)E + \frac{1}{2E}\left(V + \rho - E^2\right) + \frac{1}{SE}\sigma_d^2.$$

(13)

The derivation of incorporating non-self-regulating species is similar to cases of heterogeneous self-regulation strengths, however, the expressions of reactivity are a little more complicated, please see Supplementary Note 6 for detailed derivation and theoretical expressions of reactivity.

**Reporting summary**
Further information on research design is available in the Nature Portfolio Reporting Summary linked to this article.

## Data availability
All data analysed (except those in Fig. 3d) are simulation data and can be reproduced by using the codes provided. Data analysed in Fig. 3d are publicly available and can be found in the corresponding references[49,51–53].

## Code availability
All source codes related to our work can be found at[68] https://github.com/Pawn053/Reactivity-of-complex-ecosystems. The codes are written using MathWorks MATLAB R-2020b.

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

## Acknowledgements

We thank the members of the Li Lab for helpful comments and discussions. We gratefully acknowledge the support from the National Key Research and Development Program of China under grant no. 2022YFA1008400, the National Natural Science Foundation of China (NSFC) under grant no. 62173004, and Beijing Nova Program, China under grant no. Z211100002121105. K.R.F. is supported by Wellcome Trust Investigator award 209397/Z/17/Z and by European Research Council Grant 787932. K.Z.C. is supported by a University of Manchester Presidential Research Fellowship.

## Author contributions

Y.Y. and A.L. initiated and conceived the study. Y.Y. designed the study, performed theoretical analysis and numerical simulations under the direction of A.L. K.Z.C., K.R.F., and A.L. designed the study, performed the research, and analysed the results. Y.Y. and A.L. wrote the first draft of the manuscript, and all authors edited the manuscript.

## Competing interests

The authors declare no competing interests.
