## [Peer Review File · Nature Communications]

Reactivity of complex communities can be more important
than stabilityEditorial Note: Parts of this Peer Review File have been redacted as indicated to remove third-party material where no permission to publish could be obtained.

REVIEWER COMMENTS

Reviewer #1 (Remarks to the Author):

In "Reactivity in complex ecosystems: when short-term dynamics are more important than stability" authors show that short-term dynamics can be a better predictor of ecological outcomes than stability. They develop a theory that enables us to understand and predict how a large complex ecosystem will behave immediately after perturbations. Research reveals that many communities are reactive, so that perturbations will amplify and generate a response that is directly opposite to that predicted by typical stability measures. Reactivity is found to be prevalent for communities of mixed interactions and for structured communities.

Taken together, the results suggest that alongside stability, reactivity could be a fundamental measure for assessing ecosystem health.

Overall, I have very much enjoyed reading this paper. I find it comprehensive and clearly written, and introducing new, timely, and very interesting results that will surely also inspire future research along these lines, for example, as the authors note, for improving our understanding of ecosystem health.

I am in principle in favor of publication, but subject to the following minor revisions. The introduction falls somewhat short of giving credit to related preceding research where stability has been studied recently, for example evolutionary stability in higher-order networks, studied in Evolutionary dynamics of higher-order interactions in social networks, Unai Alvarez-Rodriguez, et al., Nat. Hum. Behav. 5, 586-595 (2021). And in terms of tipping points, physics in general, as the main branch of science responsible for the main theories used in this concept, is not really acknowledged. Seminal works here should be looked up for phase transition theory and bifurcation analysis, which are both backbones of tipping

point theory used in other fields, such as in ecology, biology, climate change, and so on.

Also, it would be very useful if the authors would make their source code available as supplementary material. This would promote the usage of the proposed theory and allow also others to take better advantage of this research, and also allow them to reproduce the results. A bit more details with regard to the simulation details, in particular for the food webs in Fig. 4 would also be most welcome.

Apart from this, I am happy to congratulate the authors on an inspiring work.

Reviewer #2 (Remarks to the Author):

The authors revisit the notion of reactivity in model ecosystems.

This notion, introduced by Neubert and Caswel in the 90s, is based on the study of linearised dynamical models near an equilibrium (which takes the form of a Jacobian matrix). A stable-reactive system is such that there exists a perturbation whose amplitude is first amplified by the dynamics before being eventually absorbed. This notion can be compared to asymptotic stability which is the long-term rate of return to the unperturbed state.

This potential amplification ought to be problematic for real ecosystems -the authors indeed show that extinction probability under repeated perturbations is sometimes more linked to reactivity than to asymptotic stability. This poses the question of whether or not ecological communities tend to be reactive. The authors here find, by randomly generating Jacobian matrices with some added structure, that there is a hierarchy for this propensity.

Exploitative (+/-) systems are the least prone to be reactive, less than competitive (-/-) and even less so than mutualistic (+/+) systems, which are the most prone to be reactive. The authors also extend the analysis to food webs, showing that the topological structure has a substantial importance on the propensity to be reactive-stable.

All in all, although similar to the work of Alesina and Tang, this a nice contribution that reads well and that ought to reinvigorate the study of transient dynamical response to perturbations in ecological systems. I must say however, that the authors have missed some

body of work that I believe to quite relevant. I am not suggesting that what the authors did has already been done, but rather that the connection needs to be made in order to build a coherent corpus. It's a bit awkward to mention my own work in this context but in (Arnoldi et al 2018 JTB) we proposed a whole theory on transient dynamical responses, very much related to what the authors study. We show that the standard notion of reactivity is not as relevant as one may think, especially in high dimensional systems, and when species have a broad range of abundances. This is because reactivity only speaks of a specific worst case perturbation, but also because rare 'satellite' species will tend to control such eigenvalues without much relevance to observable dynamics. The reason this problem doesn't show up in the present work is that the authors, following May's approach, do not explicitly consider species abundances when generating a Jacobian matrix. This issue is discussed at length in the Arnoldi et al and also in Lewi Stone(Natcomm 2020). I think this an important point to recognise if one wants to propose reactivity as a relevant notion for real ecosystems, where species abundances are typically broadly distributed.

Appart for this, I find no technical issue with the results and reiterate that the paper reads very well and contains valuable insights.

Sincerely,
JF Arnoldi.

Reviewer #3 (Remarks to the Author):

In mathematical models of large complex ecosystems, the long-term fate of a small perturbation away from an equilibrium state is generally governed by the rightmost eigenvalue of the community matrix M . Since 1972 there has grown a very large body of work in the Random Matrix Theory (RMT) literature with well established techniques to calculate this value for a variety of ecosystem models. In 1997 Neubert and Caswell pointed out that the medium-term dynamics of perturbations may be better captured by the rightmost eigenvalue of the symmetrized matrix $M+M^T$, dubbed the "reactivity". In 2014 Tang and Allesina had the idea to apply RMT results to reactivity, showing that ecosystems typically become reactive before losing stability.

In my reading the submitted manuscript has three main components:

1) The authors extend the results of Tang and Allesina to a few more cases, in particular by varying the sign pattern of the matrix to consider different types of interactions

(competitive, mutualistic, exploitative). This part is a very straightforward application of known results (references are given correctly in the methods) to simple models that are themselves not new. I do not think meets the novelty threshold for publication in Nature Communications.

2) They present some results from an empirical dataset from the mouse gut microbiome. It is quite unusual for RMT papers to make contact with real data so I was interested to read about this. However, the treatment appears to be very shallow (limited to one rather uninformative figure) and I did not get the impression anything new has been learned about the mouse gut microbiome.

3) They have conducted some simulation studies in which model ecosystems are perturbed (either once, or repeatedly), showing that "reactive" systems experience more species loss. Again this is potentially interesting, but the analysis is superficial and there is very little theory presented, so the quality of evidence is not up to the standard I would expect for Nature Communications.

In summary I do not think the manuscript meets the threshold of either novelty or scientific rigour for publication in Nature Communications. It is not a bad paper though and certainly could be published in a less selective venue.

Response to Reviewer 1

In "Reactivity in complex ecosystems: when short-term dynamics are more important than stability" authors show that short-term dynamics can be a better predictor of ecological outcomes than stability. They develop a theory that enables us to understand and predict how a large complex ecosystem will behave immediately after perturbations. Research reveals that many communities are reactive, so that perturbations will amplify and generate a response that is directly opposite to that predicted by typical stability measures. Reactivity is found to be prevalent for communities of mixed interactions and for structured communities.

Taken together, the results suggest that alongside stability, reactivity could be a fundamental measure for assessing ecosystem health.

Overall, I have very much enjoyed reading this paper. I find it comprehensive and clearly written, and introducing new, timely, and very interesting results that will surely also inspire future research along these lines, for example, as the authors note, for improving our understanding of ecosystem health.

Thank you for your positive assessment of our work and for your helpful suggestions. Below we provide point-by-point responses to each of your comments and suggestions in order.

I am in principle in favor of publication, but subject to the following minor revisions. The introduction falls somewhat short of giving credit to related preceding research where stability has been studied recently, for example evolutionary stability in higher-order networks, studied in Evolutionary dynamics of higher-order interactions in social networks, Unai Alvarez-Rodriguez, et al., Nat. Hum. Behav. 5, 586-595 (2021). And in terms of tipping points, physics in general, as the main branch of science responsible for the main theories used in this concept, is not really acknowledged. Seminal works here should be looked up for phase transition theory and bifurcation analysis, which are both backbones of tipping point theory used in other fields, such as in ecology, biology, climate change, and so on.

This is a good suggestion to make the links to related topics including evolutionary stability in higher-order networks, tipping points, phase transitions and bifurcation analysis. We have revised the first paragraph of our Introduction as follows:

L30-38: "... Stability is a general and important property of many real-world systems and is not limited to ecology. In evolutionary biology [15, 16], for example, stability analysis can be used to predict the outcomes of natural selection in diverse settings, including social networks [16]. In engineering [17, 18], stability is a key concept in the control theory of many systems, including power grids [18]. And starting in physics and bifurcation analysis, the concept of tipping points has proved useful in multiple disciplines where a loss of stability can be indicative of regime shifts and phase transitions in real-world systems [19-22]. As such, the study of stability can play an important role in predicting the occurrence of critical transitions, which is a priority in a range of contexts from financial markets to climate change [19-22]."

Also, it would be very useful if the authors would make their source code available as supplementary material. This would promote the usage of the proposed theory and allow also others to take better advantage of this research, and also allow them to reproduce the results.

We have uploaded our source code to Github (<https://github.com/Pawn053/Reactivity-of-complex-ecosystems>), and the corresponding website is provided in the revised main text (Data availability).

A bit more details with regard to the simulation details, in particular for the food webs in Fig. 4

would also be most welcome.

We now add more details about our numerical simulations at several points in the revised manuscript (e.g., several newly-added subsections in Methods section) and Supplementary Information (e.g., Supplementary Note 2, Supplementary Note 5). Specifically for food webs in Fig. 4, we have added the following details in the methods:

L408-415: "In simulations, critical values for phase transitions (i.e., the boundaries between regions with different colours in Fig. 4b) are determined as follows, taking the transition from a non-reactive state to a reactive state as an example. First, we identify the average interspecies interaction value σ_1 , at which all simulated communities are non-reactive, such that when $\sigma > \sigma_1$, not all simulated communities are non-reactive. Next, we identify the average interspecies interactions value σ_2 , at which all simulated communities are reactive, such that when $\sigma < \sigma_2$, not all simulated communities are reactive. The critical σ_r is then estimated as $\sigma_r = (\sigma_1 + \sigma_2)/2$. Similarly, we can estimate the critical σ for transition from a stable state to an unstable state."

And also flag these in the Figure 4 caption:

"The construction methods for these two food web models are provided in Methods."

Apart from this, I am happy to congratulate the authors on an inspiring work.

Thank you again for your review and very helpful suggestions.

Response to Reviewer 2

The authors revisit the notion of reactivity in model ecosystems.

This notion, introduced by Neubert and Caswel in the 90s, is based on the study of linearised dynamical models near an equilibrium (which takes the form of a Jacobian matrix). A stable-reactive system is such that there exists a perturbation whose amplitude is first amplified by the dynamics before being eventually absorbed. This notion can be compared to asymptotic stability which is the long-term rate of return to the unperturbed state.

This potential amplification ought to be problematic for real ecosystems -the authors indeed show that extinction probability under repeated perturbations is sometimes more linked to reactivity than to asymptotic stability. This poses the question of whether or not ecological communities tend to be reactive. The authors here find, by randomly generating Jacobian matrices with some added structure, that there is a hierarchy for this propensity. Exploitative (+/-) systems are the least prone to be reactive, less than competitive (-/-) and even less so than mutualistic (+/+) systems, which are the most prone to be reactive. The authors also extend the analysis to food webs, showing that the topological structure has a substantial importance on the propensity to be reactive-stable.

All in all, although similar to the work of Alesina and Tang, this a nice contribution that reads well and that ought to reinvigorate the study of transient dynamical response to perturbations in ecological systems.

Thank you for this positive summary of our work and for your helpful comments. Below we provide a point-by-point response to each comment in turn.

I must say however, that the authors have missed some body of work that I believe to quite relevant. I am not suggesting that what the authors did has already been done, but rather that the connection needs to be made in order to build a coherent corpus. It's a bit awkward to mention my own work in this context but in (Arnoldi et al 2018 JTB) we proposed a whole theory on transient dynamical responses, very much related to what the authors study. We show that the standard notion of reactivity is not as relevant as one may think, especially in high dimensional systems, and when species have a broad range of abundances. This is because reactivity only speaks of a specific worst case perturbation, but also because rare 'satellite' species will tend to control such eigenvalues without much relevance to observable dynamics. The reason this problem doesn't show up in the present work is that the authors, following May's approach, do not explicitly consider species abundances when generating a Jacobian matrix. This issue is discussed at length in the Arnoldi et al and also in Lewi Stone (Natcomm 2020). I think this an important point to recognise if one wants to propose reactivity as a relevant notion for real ecosystems, where species abundances are typically broadly distributed.

Thank you for sharing these interesting papers with us. We agree that reactivity may only provide partial information about system dynamics after perturbations. We also agree with your statement that rare 'satellite' species can affect the map between system reactivity and the dynamics after perturbation.

Prompted by your comments and prior work, we explored this phenomenon in our own modelling framework, using the generalised Lotka-Volterra models to numerically simulate removing either an abundant species or then a rare 'satellite' species. Specifically, we modelled communities with \$S - 1\$ abundant species who each have a unit equilibrium abundance, and 1 satellite species which has abundance \$X_s^* \ll 1\$. We systematically varied the interactions within these communities, capturing random, exploitative, competitive, and cooperative systems. We find that regardless of interaction type, the system yields almost the same reactivity value before and after removing a random abundant species, however, the system yields different reactivity values before

and after removing the rare satellite species (see Supplemental Fig. S15 attached below). This confirms your statement that rare species can influence a system’s reactivity value, even if they do not influence the dynamics in practice (see Supplemental Fig. S16 attached below).

We agree that this is an important point and we now cite your work, add a new paragraph in the Discussion section, add a new note in Supplementary Information (Supplementary Note 7), and add two supplemental figures (Supplemental Figs. S15-S16) into our revised manuscript. The new paragraph in the Discussion section is as follows:

L313-327: “The ability of reactivity as a measure to predict short-term dynamics can be lessened in some contexts. The initial amplification rate strongly depends on perturbation directions, and thus being reactive does not imply that all perturbations can be amplified initially [29]. Another context in which reactivity as a measure can be decoupled from short-term dynamics is when there is a wide distribution of species abundances. Arnoldi et al. studied the influence of rare species and found that rare species tend to influence measures of system stability even though they do not greatly affect system dynamics in practice [29]. In the same vein, we explored such influence of rare species on our measure of system reactivity within a range of different community types (see Supplementary Note 7 and Fig. S15). This work shows that regardless of community type, rare species can again influence system reactivity values, even though again these species would not in practice have a large impact on system dynamics (Fig. S16). To analyse such cases and gain a comprehensive view on short-term system dynamics after perturbations, one should consider other measures, such as median return rate [29] proposed by Arnoldi et al., along with system reactivity and stability. However, because our main analyses did not include such rare species, the measure of reactivity we have employed should map well to short-term dynamics.”

Supplemental Fig. S15. Rare species influence system reactivity measures. Circles show the reactivity of original system with $S - 1$ abundant species and 1 rare satellite species. Crosses show the system reactivity after randomly removing an abundant species. Diamonds show the system reactivity after removing the rare satellite species. Each symbol (circle, cross or diamond) is an average of 50 randomly-generated communities with the same community parameters

combination. The original system contains $S - 1$ abundant species with unit abundance and 1 rare satellite species. The abundance of the satellite species is shown in the title of each panel. Community parameters are: $S = 100$, $C = 0.2$, $d = 1$, $\sigma = 0.05$.

Supplemental Fig. S16. Rare species have little influence on the observable short-term dynamics. Here we use average return rate over the interval $[0, 3]$ as a representative of observable short-term dynamics. Note that the system is perturbed at time 0. Circles show the average $\mathcal{R}_3^{\text{avg}}$ for the original system with $S - 1$ abundant species and 1 rare satellite species. Crosses show the average $\mathcal{R}_3^{\text{avg}}$ after randomly removing an abundant species. Diamonds show the average $\mathcal{R}_3^{\text{avg}}$ after removing the rare satellite species. Each symbol (circle, cross or diamond) is an average of 50 randomly-generated communities with the same community parameters combination. The perturbation strength is 1 and is randomly distributed on each species. Community parameters are the same as those in Supplemental Fig. S15.

Appart for this, I find no technical issue with the results and reiterate that the paper reads very well and contains valuable insights.

Sincerely,
JF Arnoldi.

Thank you again for reviewing our paper so carefully and providing these helpful and constructive comments.

Response to Reviewer 3

In mathematical models of large complex ecosystems, the long-term fate of a small perturbation away from an equilibrium state is generally governed by the rightmost eigenvalue of the community matrix M . Since 1972 there has grown a very large body of work in the Random Matrix Theory (RMT) literature with well established techniques to calculate this value for a variety of ecosystem models. In 1997 Neubert and Caswell pointed out that the medium-term dynamics of perturbations may be better captured by the rightmost eigenvalue of the symmetrized matrix $M+M^T$, dubbed the "reactivity". In 2014 Tang and Allesina had the idea to apply RMT results to reactivity, showing that ecosystems typically become reactive before losing stability.

Thank you for your careful review and for your helpful and constructive comments. Below we provide a point-by-point response to your concerns.

In my reading the submitted manuscript has three main components:

1) The authors extend the results of Tang and Allesina to a few more cases, in particular by varying the sign pattern of the matrix to consider different types of interactions (competitive, mutualistic, exploitative). This part is a very straightforward application of known results (references are given correctly in the methods) to simple models that are themselves not new. I do not think meets the novelty threshold for publication in Nature Communications.

We fully agree that our work builds on the important results of Tang and Allesina. However, we also believe our results are still an important development on these results, because here for the first time we develop a new analytic expression that allows us to comprehensively map the reactivity of any arbitrary community type, containing any mixture of interaction types. This new theory, and our numerical extension to structured networks, allows us to systematically compare the reactivity of many possible community types, and capture the interaction distributions of real communities in a way that has not been done before. This, in turn, allows us to uncover several important new results: 1) That communities containing mixtures of interactions, or interaction structure (as we expect in most ecosystems) are typically reactive, meaning that reactivity is likely to be widespread within nature. 2) That cooperative systems, often thought to be the least stable, are in fact the least reactive. This counterintuitive result arises because, while cooperation can generate reactivity, it only really occurs in unstable systems that would not persist anyway. 3) That reactive communities are at high risk of suffering stochastic species extinctions when exposed to external perturbations, even if they are linearly stable, meaning reactivity is likely to play a critical role in overall community dynamics.

In demonstrating that reactivity is likely to be widespread, and linking it to a key measure of ecosystem health, we do believe our work will be of interest to a wide audience.

Nonetheless, prompted by your comments, we have now extended our analyses in a number of ways to make additional novel contributions. In particular, we now provide further analysis of the influence of food web structure on reactivity, which again shows that reactivity is widely predicted, further bolstering our key conclusions. We also study the effects of heterogeneity in species self-regulation upon reactivity, which reveals that reactivity of a system can be greatly increased if even just one species has limited self-regulation. Such heterogeneity in self-regulation is again expected in real communities, and so again support our key conclusion that reactivity is common. We also now add new analyses of real data which support the ability of our framework to correctly predict reactivity. Finally, we provide a new, more systematic analysis of the link between reactivity and species extinctions under perturbation, which further bolsters our finding that reactivity is not only common but also important for species loss.

We now go through these new additions one-by-one and respond to your other comments as we go.

Analyzing the influence of food web structure on system reactivity

Many natural ecological networks are structured, and a key question is how this structure affects the properties of a given community. For this reason, in the previous version of the MS, we presented work on the influence of cascade food web and niche food web on system reactivity. As is shown by Fig. 4b, we find that compared with unstructured cases, cascade food webs have no influence on reactivity, while niche food webs make systems more reactive. This work then predicted that reactivity will be common in structured communities, and that some structures will promote reactivity further than expected from an unstructured case.

However, what was lacking from this analysis was a detailed understanding of how structure can affect reactivity. We now provide this, which reveals that it is the degree distribution of the associated networks that is the main driver for changes in reactivity with structure. We show this by analysing three variants of the cascade model (interval cascade model, cascade model with broad degree distribution and interval cascade model with broad degree distribution). This allows us to generalise from our previous results and, as discussed, lend further support to our prediction that reactivity is common.

To go through this new work, we have added a new paragraph in the manuscript, a new panel in Fig. 4, a Supplementary Note (Supplementary Note 5), and three new Supplemental Figures (Supplemental Fig. S5-S7).

The new paragraph is as follows:

L229-247: “Our work shows that, as compared with an unstructured equivalent, cascade food webs do not influence system reactivity, while a niche food web structure makes the system more reactive (Fig. 4b). We find that this difference between the two food web types is explained by the different structural features introduced by each. The cascade model introduces trophic levels, which only influence the arrangement of pairwise positive and negative elements in the community matrix \mathbf{M} (the non-zero negative elements are in the upper-triangular part, while the non-zero positive elements are in the lower-triangular part, Fig. 4a). This single feature is mitigated in $\mathbf{H} = (\mathbf{M} + \mathbf{M}^T)/2$, and, as a result, the cascade model and its unstructured counterpart yield the same \mathbf{H} , and therefore the same reactivity. The niche model is different in that it introduces intervality (that is, each predator consumes preys that are adjacent in the hierarchy) and a broader degree distribution (Fig. 4c, degree of a species is the number of its interacting species) alongside introducing trophic levels. These properties not only change the arrangement of pairwise positive and negative elements in the community matrix but also the topological structure of the underlying interaction network. This change of topological structure influences matrix \mathbf{H} . Thus, $\mathbf{H}_{\text{niche}}$ is different from $\mathbf{H}_{\text{unstructured}}$, which leads to a more reactive community. Further analysis of some variants of cascade model (interval cascade model, cascade model with broad degree distribution, and interval cascade model with broad degree distribution, see Supplementary Note 5 and Figs. S5-S7) shows that the broader degree distribution is the main contributor to the reactivity increase. On this basis, we can conclude that the degree distribution is the key structural feature that drives increased reactivity.”

Fig. 4. Structured food webs display reactivity. **a**, Schematic illustration of cascade and niche food webs. In the cascade model, trophic levels are introduced, i.e., species with higher ranks predate species with lower ranks with a fixed probability. In the niche model, species predate a subset of species within a defined range or niche (shaded circles with different colours). The corresponding community matrices are presented with positive (blue) and negative (red) interactions. The construction methods for these two food web models are provided in Methods. **b**, Phase transitions of cascade and niche food webs. The blue region marks stable and non-reactive communities, red region marks stable and reactive communities, and grey region marks unstable communities. Red (blue) hollow dots are the percentage of reactive (stable) communities from numerical simulations, obtained from 50 randomly-constructed communities for each point in parameter space. For comparison, the vertical red (blue) dashed lines show the critical average interspecies interaction strength (i.e., σ) for the transition to instability (reactivity) of the corresponding unstructured food webs. As σ increases, these two food webs also experience two phase transitions: one is from stable non-reactive state to stable reactive state, the other is from stable reactive state to unstable state. The cascade model shares the same critical σ with the unstructured case to move into a stable reactive state, while the niche model requires a lower average strength of interspecies interaction to move into a stable reactive state. **c**, Degree distribution for these three types of food webs. In this figure, community parameters are: $S = 500$, $C = 0.3$, $\sigma = 0.1$.

Supplemental Fig. S5. Reactivity of different variants of cascade model. Variant 1 is interval cascade food web, variant 2 is cascade food web with broad degree distribution, and variant 3 is interval cascade food web with broad degree distribution. Red lines represent the reactivity of the original cascade model, grey symbols represent the reactivity values of different variants of cascade model. For each type of variant, we randomly generate 50 communities with the same community parameters combination and plot their reactivity. In this figure, we have $S = 500$, $C = 0.2$, $\sigma = 0.15$, $d = 1$.

Supplemental Fig. S6. Reactivity change of variants of cascade model. Each data in the histogram is an average of 1000 randomly generated communities with the same set of parameters. In this figure, we set $\sigma = 0.15$ and $d = 1$.

Supplemental Fig. S7. Degree distributions of different food web models. The parameters in this figure are the same as Supplemental Fig. S5.

Self-regulation and reactivity of complex ecosystems

Like structuring, species self-regulation – and particular, the heterogeneity of self-regulation – is considered a key determinant of ecosystem dynamics. However this feature has been relatively understudied in relation to reactivity. In our revision, we now extend our theoretical framework to allow both heterogeneous self-regulation strengths and non-self-regulating species, providing a valuable new comprehensive study of how these features influence system reactivity (Supplementary Note 6).

This new analysis reveals that increasing the heterogeneity of self-regulation strengths makes the system more reactive (see Fig. 5a and Supplemental Fig. S10 attached below) and, surprisingly, just a single species without self-regulation is enough to switch the system from non-reactive to reactive (see Fig. 5c and Supplemental Fig. S10 attached below). We further find that these two conclusions are robust to the introduction of trophic levels (see Fig. 5b and Supplemental Fig. S10 attached below). As discussed above, this analysis further supports the key overarching conclusion of our work: that reactivity is likely to be common within real ecosystems.

We discuss these new results with a section in the Results, a new paragraph in the Discussion section, details of the approach in the Methods section, a new Supplementary Note (Supplementary Note 6) and three new Supplemental figures.

The new section in the Results is as follows.

L248-284: “Self-regulation and reactivity of complex ecosystems. A key factor known to influence community stability is the strength of density dependent regulation exhibited by the populations of each species in a community [3, 4, 11], also known as self-regulation (i.e., M_{ii}). Broadly speaking, increased self-regulation is stabilising and, therefore, is predicted to shift communities to a more stable state [4]. As might be expected, therefore, it is also the case that increasing self-regulation can lead to a less reactive community (since it is clear from the theoretical expressions that more negative M_{ii} , i.e., larger self-regulation strength, can bring a smaller reactivity value). However, our model assumes that all species have identical levels of self-regulation. In practice, the expectation [11, 47, 48] is that there will be variability in self-regulation between species with the possibility of some species that have very limited self-regulation, which are instead regulated by their interactions with the wider community. We, therefore, sought to understand the potential impacts of such self-regulation heterogeneity on system reactivity, and whether a system can remain non-reactive when some species do not self-regulate. Here, we were able to leverage recent developments [11, 49-51] of random matrix theory in order to incorporate heterogeneous self-regulation strengths and non-self-regulating-species into our theoretical framework of system reactivity (Methods and Supplementary Note 6).

With this method in place, we can vary the heterogeneity of self-regulation in unstructured networks as the standard deviation of self-regulation strength σ_d , which reveals that increasing self-regulation heterogeneity makes the system more reactive (Fig. 5a and Fig. S10, see Supplementary Note 6 for theoretical analysis). We next ask whether the introduction of network structure alters this prediction, using the case of a cascade food web. Specifically, here we consider three cases. In the first, the strengths of self-regulation have no relation to trophic level (disorganised case). The second assumes that higher trophic level possesses higher self-regulation strength (positive case). In the third, species with higher trophic level possesses lower self-regulation strength (negative case). We find that all these three cases yield the same result as unstructured food web (Fig. 5b). Our findings on the importance of heterogeneity in self-regulation for reactivity, therefore, are robust to these changes in network structure.

Similarly, we can explore the influence of non-self-regulating species on system reactivity in communities of otherwise homogeneous self-regulation strengths. Strikingly, we find that

even a single species without self-regulation is enough to make a non-reactive system reactive, regardless of the self-regulation strength of other community members (Fig. 5c, Fig. S10, see Supplementary Note 6 for theoretical analysis). For most community types, this removal of self-regulation typically also renders communities unstable. However, we find that exploitative systems are able to remain stable even when multiple species no longer self-regulate, yet become increasingly reactive as the number of non-self-regulating species increases. This finding holds in the presence of trophic levels (Fig. S10), and when those species that do self-regulate do so in a heterogeneous manner (Fig. S10). Together, these analyses suggest that all species must self-regulate with relatively high strengths in order to prevent community reactivity.”

The newly added paragraph in Discussion is as follows.

L328-335: “We also find that the degree of self-regulation by individual species in a community can be important for system reactivity. Self-regulation has long been known to have the potential to promote system stability, with some studies suggesting that the majority of species need to have strong self-regulation strengths to keep stability [11]. Our work suggests that to prevent reactivity the requirements for self-regulation are even stricter, with a requirement that all species must self-regulate with relatively high strengths. Given we also find conditions where reactivity drives species extinctions, this finding suggests that self-regulation may play a more important role in ecosystem health than previously appreciated.”

We have added the following to the Methods, and also include a lengthier derivation in our revised Supplementary Information (Supplementary Note 6):

L431-445: “**Constructing community matrices for communities with heterogeneous self-regulation strengths.** The only difference between the community matrices of communities with homogeneous self-regulation strengths and communities with heterogeneous self-regulation strengths is the diagonal terms (i.e., M_{ii}), and the sampling of off-diagonal terms is the same as presented previously. Thus, here we focus solely on the sampling of diagonal terms. When there is no interplay between self-regulation strengths and trophic levels, all diagonal terms are sampled from a uniform distribution $U[-d_{\text{mean}} - \sqrt{3}\sigma_d, -d_{\text{mean}} + \sqrt{3}\sigma_d]$ respectively and independently. Here d_{mean} is the average self-regulation strength. When there exists an interplay between self-regulation strengths and trophic levels, we first generate S random values d_1, \dots, d_S from the uniform distribution $U[-d_{\text{mean}} - \sqrt{3}\sigma_d, -d_{\text{mean}} + \sqrt{3}\sigma_d]$ respectively and independently. When species with higher trophic level possess higher self-regulation strength, we sort d_i in an ascending order and set $M_{ii} = -d_i$ (note that i indicates the trophic level). When species with a higher trophic level possess lower self-regulation strength, we sort d_i in a descending order and set $M_{ii} = -d_i$. Note that this construction method can be adapted to cases where self-regulation strengths are sampled from other distributions.”

L446-452: “**Constructing community matrices for communities with non-self-regulating species.** Here we still focus on the determination of M_{ii} . When there is no interplay between self-regulation strengths and trophic levels, we first generate N distinct positive integers less than or equal to S randomly: $Q = \{q_1, \dots, q_N\}$. For each $i \in Q$, set $M_{ii} = 0$. For each $i \notin Q$, set $M_{ii} = -d$. When species with higher trophic level self-regulate, we set $M_{ii} = 0$ for $i \leq N$ and $M_{ii} = -d$ for $i > N$ (note that i indicates the trophic level). When species with higher trophic level do not self-regulate, we set $M_{ii} = -d$ for $i \leq S - N$ and $M_{ii} = 0$ for $i > N$.”

L453-474: “**Incorporating heterogeneous self-regulation strengths and non-self-regulating species into reactivity analysis.** For convenience, we still denote $E(M_{ij})_{i \neq j} =$

E , $\text{Var}(M_{ij})_{ij} = V$, and $E(M_{ij}M_{ji})_{i \neq j} = \rho$, which leads to the statistics of matrix \mathbf{H} : $E(H_{ij})_{i \neq j} = E$, $\text{Var}(H_{ij})_{ij} = (V + \rho - E^2)/2$, and $E(H_{ij}H_{ji})_{i \neq j} = (V + \rho + E^2)/2$.

First, we discuss cases of heterogeneous self-regulation strengths. For the simplicity of theoretical derivation, here we focus on the case where self-regulation strengths are sampled from a uniform distribution $[-d_{\text{mean}} - \sqrt{3}\sigma_d, -d_{\text{mean}} + \sqrt{3}\sigma_d]$. For the classic random community, we have $E = 0$, $V = C\sigma^2$ and $\rho = 0$. We can then identify the rightmost eigenvalue of \mathbf{H} (see Supplementary Note 6 for detailed derivation), which leads to the expression of reactivity

$$\mathcal{R} = \frac{\sqrt{2}}{2} \sigma \sqrt{SC} \left(\frac{d_1 + d_2}{2} + \sqrt{\left(\frac{d_2 - d_1}{2}\right)^2 + 1} + \frac{2}{d_2 - d_1} \tanh^{-1} \left(\frac{d_2 - d_1}{\sqrt{(d_2 - d_1)^2 + 4}} \right) \right),$$

where

$$\begin{aligned} d_1 &= (-\sqrt{2}d_{\text{mean}} - \sqrt{6}\sigma_d) / \sigma \sqrt{SC}, \\ d_2 &= (-\sqrt{2}d_{\text{mean}} + \sqrt{6}\sigma_d) / \sigma \sqrt{SC}. \end{aligned}$$

For communities with mixed interaction types, when $|E| < \sqrt{(V + \rho - E^2)/(2S) + \sigma_d^2/S^2}$, we can identify the rightmost eigenvalue of \mathbf{H} (see Supplementary Note 6 for detailed derivation), which leads to the expression of reactivity

$$\begin{aligned} \mathcal{R} &= \sqrt{\frac{1}{2}S(V + \rho - E^2)} \\ &\cdot \left(\frac{d_1 + d_2}{2} + \sqrt{\left(\frac{d_2 - d_1}{2}\right)^2 + 1} + \frac{2}{d_2 - d_1} \tanh^{-1} \left(\frac{d_2 - d_1}{\sqrt{(d_2 - d_1)^2 + 4}} \right) \right) \\ &- E \end{aligned}$$

where

$$\begin{aligned} d_1 &= (-d_{\text{mean}} - \sqrt{3}\sigma_d) / \sqrt{\frac{1}{2}S(V + \rho - E^2)}, \\ d_2 &= (-d_{\text{mean}} + \sqrt{3}\sigma_d) / \sqrt{\frac{1}{2}S(V + \rho - E^2)}. \end{aligned}$$

When $|E| > \sqrt{(V + \rho - E^2)/(2S) + \sigma_d^2/S^2}$, the expression of reactivity can be derived as follows (see Supplementary Note 6 for detailed derivation)

$$\mathcal{R} = \max(\mathcal{R}_1, \mathcal{R}_2),$$

where

$$\begin{aligned} \mathcal{R}_1 &= \sqrt{\frac{1}{2}S(V + \rho - E^2)} \\ &\cdot \left(\frac{d_1 + d_2}{2} + \sqrt{\left(\frac{d_2 - d_1}{2}\right)^2 + 1} + \frac{2}{d_2 - d_1} \tanh^{-1} \left(\frac{d_2 - d_1}{\sqrt{(d_2 - d_1)^2 + 4}} \right) \right) \\ &- E, \end{aligned}$$

and

$$\mathcal{R}_2 = -d_{\text{mean}} + (S - 1)E + \frac{1}{2E}(V + \rho - E^2) + \frac{1}{SE}\sigma_d^2.$$

The derivation of incorporating non-self-regulating species is similar to cases of heterogeneous self-regulation, however, the expressions of reactivity are a little more complicated, please see Supplementary Note 6 for detailed derivation and theoretical

expressions of reactivity.”

The newly-added figure is as follows:

Fig. 5. Influence of self-regulation on system reactivity. **a & b**, Influence of self-regulation heterogeneity on system reactivity. Self-regulation heterogeneity is quantified by the standard deviation of self-regulation strengths. Solid lines are theoretical predictions, symbols (dots, circles, diamonds, and crosses) are results from numerical simulations. ‘Disorganised’ refers to the case where self-regulation strengths have no interplay with trophic levels, ‘positive’ refers to the case where species with higher trophic level have higher self-regulation strength, and ‘negative’ refers to the case where species with higher trophic level have lower self-regulation strength. Note that in these two panels, all communities concerned are stable. **c**, Influence of non-self-regulating species on system reactivity. Upper panels show average community state, with the blue region indicating communities are on average stable and unreactive, the red region indicating stable and reactive, and the grey region indicating unstable and reactive. Lower panel shows the average reactivity values for each parameter combination. In (a) and (b), the average self-regulation strength is $d_{\text{mean}} = 6$, and self-regulation strengths are sampled from a uniform distribution. Other community parameters in this figure are $S = 200$, $C = 0.3$, $\sigma = 0.1$. Each simulation data point is the average of 50 randomly-constructed communities for the specific parameter combination.

The newly added Supplemental figures are as follows:

Supplemental Fig. S8. Eigenvalue distributions of four community types with heterogeneous self-regulation strengths. Coloured dots are eigenvalues of H of 20 randomly generated communities. Black dots are theoretical predictions of the endpoints of the bulk of eigenvalues and the outlier.

In this figure, self-regulation strengths are sampled from a uniform distribution. Community parameters are $S = 250$, $C = 0.25$, $\sigma = 0.25$, $d_{\text{mean}} = 1$ and $\sigma_d = 1$.

Supplemental Fig. S9. Eigenvalue distributions of four community types where some species do not self-regulate. Coloured dots are eigenvalues of H of 20 randomly generated communities. Black dots are theoretical predications of the rightmost eigenvalue of matrix H . In this figure, $S = 250$, $C = 0.25$, $\sigma = 0.25$, $d = 3$ and 20% species do not self-regulate.

Supplemental Fig. S10. **a & b**, Simulated results on the influence of heterogeneous self-regulation strengths on system reactivity. **c**, Theoretical results on the influence of non-self-regulating species on system reactivity. Note that here self-regulating species have homogeneous self-regulation strengths. **d**, Simulated results on the influence of non-self-regulating species on the reactivity of cascade food webs. The term 'disorganised' refers to the case where self-regulation strengths have

no interplay with trophic levels, 'positive' refers to the case where species with higher trophic tend to self-regulate, and 'negative' refers to the case where species with lower trophic level tend to self-regulate. e, Simulated results on the influence of non-self-regulating species on system reactivity. Note that here self-regulation strengths of self-regulating species are sampled from a uniform distribution with standard deviation 0.3. In (d) and (e), Communities in the blue region are on average stable and non-reactive, communities in the red region are on average stable and reactive, while communities in the grey region are on average unstable and reactive. Each data point in the simulated results is an average of 50 randomly-constructed communities for the specific combination of community parameters. Community parameters are the same as those in Fig. 5.

2) They present some results from an empirical dataset from the mouse gut microbiome. It is quite unusual for RMT papers to make contact with real data so I was interested to read about this. However, the treatment appears to be very shallow (limited to one rather uninformative figure) and I did not get the impression anything new has been learned about the mouse gut microbiome.

Apologies, our work with real data was too cursory in the first version. We have reworked the descriptions of this part of the paper to make the methods clearer, and also expanded it to include the analysis for several additional data sets. Together, these analyses both show that our framework is able to capture the properties of real data and, moreover, that reactivity is predicted across a diverse set of real microbial communities.

Turning first then to the more extensive description of what was done in the Methods.

L370-385: "Applying the theory to experimental data. We utilize data from independent empirical studies of seven different microbial communities [36-39] to parameterise our theoretical model, and thereby assess the level of reactivity of several real communities. Specifically, for each community, the original researchers have already inferred the true interaction networks (A_{ij}) and growth rates (r_i). For a given community, we then perform the following steps: 1) Using the empirically derived A_{ij} and r_i parameters, we identify the underlying equilibrium abundances of each species within the community, then determine the empirical community matrix \mathbf{M}_e . 2) We then determine the true reactivity directly from \mathbf{M}_e , via calculating the eigenvalues of $\mathbf{H}_e = (\mathbf{M}_e + \mathbf{M}_e^T)/2$ (blue bars, Fig. 2c). 3) We then also determine the average properties of \mathbf{M}_e for each community. That is, we determine the connectance (C) and average self-regulation (d), and the size and shape of the interaction distribution (assuming interaction parameters follow a Gamma distribution, although any appropriate distribution can be used). 4) We then feed these average community properties into our theoretical framework to derive a theoretical estimate of the reactivity of each community (red dots, Fig. 2c). Comparing the red dots to the tops of the blue ribbons thus enables us to assess the ability of our theoretical analysis to accurately estimate the true reactivity of realistic community networks."

As discussed, above we now apply these procedures to additional dataset from other microbial communities (see revised Fig. 2 attached below). In all cases, we find that our theoretical predictions of reactivity match well to reactivity values drawn directly from interaction data. Moreover, in each natural community, we predict that there is a positive reactivity value, whereas we find that in a randomly-constructed community (assembled in vitro from a combination of soil microbial species and species from the *C. elegans* intestine) reactivity is not predicted. Note that the randomly-constructed community is not likely to exist in natural settings. To make all of this clearer, we have revised our discussion of these analyses and their implications in the main text, and we also provide a new Supplementary Note (Supplementary Note 2) detailing the data from the microbial communities studied, and a new Supplementary figure.

The revised part in the main text is as follows:

L131-146: “This extension enables us to estimate the level of reactivity in real ecological networks from their average network properties, provided suitable data are available. We used our framework to explore the reactivity of seven microbial communities, two found within the mammalian gut [36, 37], four isolated from the soil [38], and one arbitrarily assembled *in vitro* using a mixture of gut and soil associated microbes [39] (see Supplementary Note 2). In each case, we used our theoretical framework to estimate the reactivity of each community based solely on their average properties – that is, the average sign and strength of interspecies interactions – and compared this to the measures of reactivity derived directly from each species’ individual ecological parameters (see Methods). Interestingly, our model predicts that each natural microbial community will be reactive, but that the artificially assembled community is non-reactive. Moreover, for all the communities, our theoretical prediction of reactivity (Fig. 2c, red dots) matches to the community reactivity calculated directly from each interaction network (Fig. 2c, blue bars). Together, these tests confirm that our theory is able to predict the reactivity of real systems using only their average community properties. Moreover, these analyses provide further support for the prediction that reactivity is widely expected in real communities.”

Fig. 2. Reactivity criteria for large complex ecosystems. **a**, Predicting the eigenvalue distribution of H and corresponding reactivity profiles. In the first row, the maximum (grey circles) and minimum eigenvalues (grey diamonds) of 50 randomly generated communities are plotted. Blue and red lines are theoretical predictions of maximum and minimum eigenvalues, respectively. In the second row, we systematically vary C to obtain $\sigma\sqrt{SC}$ spanning $[0,1.5]$. Orange dots represent the percentage of reactive communities out of 100 samples from numerical simulations. Blue lines are the corresponding theoretical predictions for the numerical percentage. Grey regions show unstable communities. In all cases, phase transitions from non-reactivity to reactivity are well predicted by our theory. **b**, Reactivity criteria for different types of communities. Curves

with different colours are critical $C - S$ curves for different communities, and combinations of S and C below each curve lead to non-reactive communities. Different types of communities form a strict hierarchy from predator-prey communities (most likely to be non-reactive) to mutualistic communities (most likely to be reactive). c, We extend our theory to communities with mixed interaction types. This extended theory can predict the reactivity of empirical microbial communities. Red dots are the dominant eigenvalue predicted by our theory, blue bars are the dominant eigenvalue computed directly from experimental data. Here communities 1 and 2 are mouse microbial communities [36, 37], communities 3 to 6 are soil microbial communities [38], and community 7 is a randomly-constructed community [39]. For detailed information of empirical microbial communities studied in our paper, please see Supplementary Note 2. In (a) and (b), we have $d = 1$. In the first row of (a), we have $S = 150$, $C = 0.25$, $\sigma = 0.25$. In the second row of (a), other parameters are the same as the first row except for the varying C . In (b), we have $\sigma = 0.1$.

3) They have conducted some simulation studies in which model ecosystems are perturbed (either once, or repeatedly), showing that "reactive" systems experience more species loss. Again this is potentially interesting, but the analysis is superficial and there is very little theory presented, so the quality of evidence is not up to the standard I would expect for Nature Communications.

Thank you for this comment. With hindsight, we agree we did not provide enough breadth to these analyses. We have now expanded our analyses of the links between reactivity and extinctions by assessing extinctions under a range of different perturbations, including repeated unidirectional perturbations, repeated bidirectional perturbations and repeated stochastic perturbations (see Supplemental Fig. S11 attached below for illustration of these perturbation types). These analyses all support our key conclusion that reactive systems are more likely to suffer species loss under frequent perturbations (see Fig. 6 and new Supplemental Figs. S11-S14 attached below).

We have edited the manuscript to draw the reader's attention to all of these new analyses:

L294-297: "Now, the effects of reactivity dominate the dynamics that are observed and, importantly, reactive communities are more likely to experience species extinctions than non-reactive ones (Fig. 6a, bottom and Fig. 6b). We find this prediction to be general across a range of different perturbation types and frequencies (Figs. S11-S14)."

Newly added figures are as follows.

Fig. 6. Reactivity predicts species persistence better than stability under frequent perturbations. **a**, Responses of different food webs to perturbations. Each line in (a) captures the change in abundance of each species after the perturbations. Green regions indicate that the community recovers to its equilibrium, and red regions suggest that the community suffers species loss. Although both non-reactive stable and reactive stable communities can recover from a single perturbation (upper row of (a)), reactive stable communities can experience species loss following frequent perturbations (bottom row of (a)). **b**, Overall view of species loss in typical food webs under frequent perturbations with different strengths and frequencies. Colours represent the percentage of communities suffering species loss from numerical simulations. To obtain the corresponding percentage, we randomly construct 20 communities and count how many communities can persist under frequent perturbations. In each analysis, community dynamics are simulated using the gLV model with a species regarded as extinct if its abundance falls below 0.02, and community parameters are: $S = 50$, $C = 0.2$, $\sigma = 0.05$. Per capita self-regulation strength $s = 1$ for non-reactive stable communities, and $s = 0.1$ for reactive stable communities. Frequent perturbations are imposed by increasing the abundances of 30 randomly-picked species with a fixed frequency (we name such type of frequent perturbations as repeated unidirectional perturbation, see Fig. S9). For communities in (a), equilibrium abundance is 2 for each species. For communities in (b), equilibrium abundance is 1 for each species and simulation time is 500 units.

Supplemental Fig. S11. Different types of frequent perturbation evaluated in our paper. In the case

of repeated unidirectional perturbations, species abundances of perturbed species are increased (or decreased) by the same amplitude with a fixed frequency. In the case of repeated bidirectional perturbations, species abundances of perturbed species are increased and decreased alternatively by the same amplitude with a fixed frequency. In the case of repeated stochastic perturbation, species abundances of perturbed species are increased or decreased with a fixed frequency, and the perturbation at each time point is sampled from a random distribution (in this figure, normal distribution).

Supplemental Fig. S12. Overall view of species loss in typical food web models under repeated unidirectional perturbations. Colours represent the percentage of communities suffering species loss from numerical simulations. To obtain the corresponding percentage, we randomly construct 20 communities and count how many communities can persist under frequent perturbations. Colour close to green suggests a higher percentage, while colour close to yellow suggests a lower percentage. In this figure, community dynamics is the generalised Lotka-Volterra model. In **a**, community size $S = 10$, per capita self-regulation strength $s = 0.1$ for non-reactive stable case, and $s = 0.01$ for reactive stable case. In **b**, community size $S = 50$, per capita self-regulation strength $s = 1$ for non-reactive stable case, and $s = 0.1$ for reactive stable case. Other community parameters are the same as those in Fig. 6b. Repeated unidirectional perturbations are imposed by increasing the abundances of randomly-picked 60% species with a fixed frequency. Simulation time is 500 unit time. In the simulations, we regard a species as extinct if its abundance is less than 0.02.

Supplemental Fig. S13. Overall view of species loss in typical food web models under repeated bidirectional perturbations. Colours represent the percentage of communities suffering species loss from numerical simulations. To obtain the corresponding percentage, we randomly construct 20 communities and count how many communities can persist under frequent perturbations. Colour close to green suggests a higher percentage, while colour close to yellow suggests a lower percentage. In this figure, community dynamics is the generalised Lotka-Volterra model. In **a**, community size $S = 10$, per capita self-regulation strength $s = 0.1$ for non-reactive stable case, and $s = 0.01$ for reactive stable case. In **b**, community size $S = 50$, per capita self-regulation strength $s = 1$ for non-reactive stable case, and $s = 0.1$ for reactive stable case. Other community parameters are the same as those in Fig. 6b. Repeated bidirectional perturbations are imposed by increasing and decreasing the abundances of randomly-picked 60% species alternatively with a fixed frequency. Simulation time is 500 unit time. In the simulations, we regard a species as extinct if its abundance is less than 0.02.

Supplemental Fig. S14. Overall view of species loss in typical food web models under repeated stochastic perturbations. Colours represent the percentage of communities suffering species loss from numerical simulations. To obtain the corresponding percentage, we randomly construct 20 communities and count how many communities can persist under frequent perturbations. Colour close to green suggests a higher percentage, while colour close to yellow suggests a lower percentage. In this figure, community dynamics is the generalised Lotka-Volterra model. In **a**, community size $S = 10$, per capita self-regulation strength $s = 0.1$ for non-reactive stable case, and $s = 0.01$ for reactive stable case. In **b**, community size $S = 50$, per capita self-regulation strength $s = 1$ for non-reactive stable case, and $s = 0.1$ for reactive stable case. Other community parameters are the same as those in Fig. 6b. In this figure, abundances of perturbed species are increased or decreased with a fixed frequency, and perturbation at each time point is generated by a normal distribution with mean 0. Randomly-picked 60% species are perturbed. Simulation time is 500 unit time. In the simulations, we regard a species as extinct if its abundance is less than 0.02.

In summary I do not think the manuscript meets the threshold of either novelty or scientific rigour for publication in Nature Communications. It is not a bad paper though and certainly could be published in a less selective venue.

Thank your again for your careful review and constructive comments and suggestions. The manuscript is much improved as a result. We hope that our extensive revisions, clarifications, and the provision of a large number of new analyses and figures make it clear that our work meets the threshold for novelty at *Nature Communications*.

REVIEWER COMMENTS

Reviewer #1 (Remarks to the Author):

The authors have revised their manuscript comprehensively and with love to detail. I warmly recommend publication in present form.

Reviewer #2 (Remarks to the Author):

The authors did a lot of work to respond to all comments. I appreciate the fact that they ran simulations to check the effect of rare species, and was impressed by the new derivations that account for non uniform diagonal interaction terms.

I have only one major comment, substantiated by a figure attached.

In most community models, such as generalised lotka-volterra, the matrix M , that arises when studying near equilibrium dynamics, will always take the form $M=D(-I+A)$ where D is a positive diagonal matrix whose elements are $D_{ii}=d_i N_i$, with d_i the self-regulation of species i , and N_i its equilibrium abundance. A is a non-dimensional matrix of relative per-capita interaction strengths (note that we have some freedom in what we put in the diagonal factor D , here I write it such that the remaining interaction matrix has a uniform diagonal).

The authors use random Matrix theory (RMT) to predict reactivity. But which matrix, M or A , is best suited to be seen as a random matrix? I would argue A , because M is structured by the diagonal term that contains species abundances.

How important is this distinction? The attached figure suggest that, if A is a random matrix and abundances (also drawn at random) are not uniform, then the fact that A is reactive implies that M is reactive, but not the converse. I think that this claim is worth exploring as, in my opinion, it would help clarify the role of interactions, and species abundances in driving reactivity.

It would also make a clear connection with Gibbs et al. "Effect of population abundances on the stability of large random ecosystems." Physical Review E 98.2 (2018): 022410, where this issue is addressed but for stability, not reactivity.

Minor point:

My last point concerns the effect of repeated perturbations. I think studying the correlation between measures of variability (see Arnoldi et al, Ecollet 2019) reactivity and asymptotic stability, would be more elegant than looking at simulation model with always the same perturbation. Indeed, those measures can all be derived from the matrix M (see Arnoldi et al, JTB I 2016). Also, it can be shown that variability integrates the whole transient response so it thus providing a mathematical explanation as to why the latter may depend more on reactivity than on the asymptotic response.

[EDITORIAL NOTE: REVIEWER #2'S ATTACHED FIGURE IS REDACTED]

Reviewer #3 (Remarks to the Author):

The authors have made substantial revisions to the manuscript in response to the first round of refereeing, including adding quite a bit of new material to expand the results. My personal view is that the work as a still whole does not meet the threshold of novelty for publication in Nature Comms, however I understand that the other referees do not share this opinion and of course it is ultimately up to the editor to decide.

I need to highlight one major issue that must be addressed before publication can be considered. I believe the analysis presented in Figure 2c has serious problems. The figure is presented as a validation of the reactivity theory for empirical communities - this is a major selling point of the paper and this figure is the only place where this claim is quantified. The "analytical prediction" red dots sit perfectly on top of experimental data in a way that is highly suspicious to anyone who has worked with real data in this setting, so I have attempted to reproduce this figure. I have not been able to reproduce the figure - my calculations (using the formulae from the manuscript and the data from the references) are some way off what is reported in the figure, although numbers I get for the experimental data are close so I think I am doing the right thing. The authors must provide a complete and detailed procedure for producing the "analytical prediction" red dots, so that others can reproduce it. I think the current figure is incorrect and misrepresents the results, and the

paper cannot be published until this issues is resolved.

The reason the agreement between the theory and data here is so shocking is that the empirical communities only have a small number of species in them. Reference 38 only reported a community matrix of size 8×8 . But as everyone who understands the body of RMT theory that this paper users should know, the results are only valid in the limit of large numbers of communities. Indeed, figure 2b includes up to $S=1500$, but one has to delve into the supplement to discover that figure 2c has just a handful of species. The authors should know that the theory they use does not apply in this setting and they should not claim that it does.

Unfortunately this fits a pattern of overstating the strength of their results. For example, in the abstract it is claimed that they "develop a new body of theory to understand and predict how any large complex ecosystem will behave immediately after perturbations"; a correct statement would be that they "apply existing theory to predict how some large linear systems respond to small perturbations". I strongly recommend the authors to careful check the manuscript for other similar exaggerations.

Response to Reviewer 1

The authors have revised their manuscript comprehensively and with love to detail. I warmly recommend publication in present form.

Thank you again for your careful review, and for your insightful and helpful comments.

Response to Reviewer 2

The authors did a lot of work to respond to all comments. I appreciate the fact that they ran simulations to check the effect of rare species, and was impressed by the new derivations that account for non uniform diagonal interaction terms.

Thank you for your time reviewing our paper, and for the positive assessment of our revised manuscript. Below we provide point-by-point responses to your comments.

I have only one major comment, substantiated by a figure attached.

In most community models, such as generalised lotka-volterra, the matrix M , that arises when studying near equilibrium dynamics, will always take the form $M=D(-I+A)$ where D is a positive diagonal matrix whose elements are $D_{ii}=d_i N_i$, with d_i the self-regulation of species i , and N_i its equilibrium abundance. A is a non-dimensional matrix of relative per-capita interaction strengths (note that we have some freedom in what we put in the diagonal factor D , here I write it such that the remaining interaction matrix has a uniform diagonal).

The authors use random Matrix theory (RMT) to predict reactivity. But which matrix, M or A , is best suited to be seen as a random matrix? I would argue A , because M is structured by the diagonal term that contains species abundances.

Figure drawn by Reviewer 2

How important is this distinction? The attached figure suggest that, if A is a random matrix and abundances (also drawn at random) are not uniform, then the fact that A is reactive implies that M is reactive, but not the converse. I think that this claim is worth exploring as, in my opinion, it would help clarify the role of interactions, and species abundances in driving reactivity.

It would also make a clear connection with Gibbs et al. "Effect of population abundances on the stability of large random ecosystems." *Physical Review E* 98.2 (2018): 022410, where this issue is addressed but for stability, not reactivity.

Thank you very much for raising this interesting and important point, and for sharing us with your simulation results. Our main analyses of system reactivity obviously follow from the classic community matrix modelling framework of Robert May and developed extensively by many others since May. As you note, this approach does not explicitly consider species abundances. Moreover, as is clear from your analysis, the potential role of species abundances in structuring the community matrix \mathbf{M} can be seen when we consider a generalised Lotka-Volterra model, where $\mathbf{M} = \text{diag}(\mathbf{X}^*)\mathbf{A}$. Here $\text{diag}(\mathbf{X}^*)$ is a diagonal matrix with species equilibrium abundances on its main diagonal. If one adopts this modelling approach, our main analyses can be interpreted as communities having homogeneous equilibrium abundances. However, natural communities tend to have more complex forms of species abundance distribution, which may indeed have impacts on system reactivity. We agree then that the relationship between the reactivity of \mathbf{A} and the reactivity of the corresponding \mathbf{M} is an interesting thing to consider.

Focusing on community matrix with the form $\mathbf{M} = \text{diag}(\mathbf{X}^*)\mathbf{A}$, Gibbs *et al.* studied the relationship between the stability of \mathbf{A} and the stability of the corresponding \mathbf{M} . They found that species abundances do not affect qualitatively stability. That is, as long as \mathbf{A} is stable, \mathbf{M} corresponding to any feasible equilibrium is stable. However, they did not address reactivity.

Prompted by your comments and simulation results, we now explore this relationship in our revised manuscript by conducting further theoretical analysis and numerical simulations. Specifically, we construct an approximation theory for cases where species abundances are sampled from a uniform distribution (see Supplementary Note 8 and Supplemental Figs. S21-S22 attached below), and we perform numerical simulations by sampling species abundances from different random distributions (see Supplemental Fig. S23 attached below). Our theoretical analysis and numerical simulations yield the same result obtained by you. That is, the reactivity of \mathbf{A} implies the reactivity of \mathbf{M} , but the non-reactivity of \mathbf{A} cannot guarantee the non-reactivity of \mathbf{M} . Please note that our theory is an approximation theory, and we leave open a full formal proof of the relationship between the reactivity of \mathbf{A} and the reactivity of the corresponding \mathbf{M} .

Based on our theory and simulations, we now discuss the relationship between the reactivity of \mathbf{A} and the reactivity of the corresponding \mathbf{M} in the revised manuscript with a new paragraph in Discussion section, a new subsection in a Supplementary Note and three new Supplemental Figures.

The new paragraph in the Discussion section is as follows.

L338-L352: *“Our main analyses of system reactivity follow the classic community matrix modelling framework of May, which has been since developed by many others. Although powerful, this approach does not account for differences in population sizes among species. Recent work by Gibbs et al. extending classic work on system stability revealed that species abundances do not qualitatively affect stability in a Lotka-Volterra framework [13]. That is, as long as an interaction matrix \mathbf{A} is stable, the community matrix $\mathbf{M} = \text{diag}(\mathbf{X}^*)\mathbf{A}$ will also be stable for any feasible equilibrium \mathbf{X}^* . This result led us to wonder whether similar rules might apply to reactivity. To investigate the potential role of species abundances on reactivity, we conducted additional theoretical analyses and numerical calculations (Supplementary Note 8, Figs. S21-S23). While our theory is only an approximation (Supplementary Note 8), it fits well with our numerics (Figs. S21 and S22) and suggests that, as with system stability, the reactivity of \mathbf{A} implies the reactivity of corresponding \mathbf{M} but the non-reactivity of \mathbf{A} cannot guarantee the non-reactivity of corresponding \mathbf{M} (Fig. S23; Supplementary Note 8). This result is important in that it implies that accounting for variability in species abundances will only increase the scope for reactivity beyond that which we have predicted here.”*

The newly-added Supplemental Figures are as follows:

Supplemental Fig. S21. Rightmost eigenvalues of H of four classic types of communities where species abundances are sampled from a uniform distribution. Grey dots are rightmost eigenvalues of H of 100 randomly generated communities. Black dots represent theoretical approximations of the rightmost eigenvalue of matrix H . In this figure, $S = 200$, $C = 0.3$, $\sigma = 0.1$, $s = 5$. Species abundances are sampled from a uniform distribution with $\mu_x = 2$, $\sigma_x = 0.1$.

Supplemental Fig. S22. Approximation of the reactivity of communities where species abundances are sampled from a uniform distribution. Dots in these panels are results from numerical simulations (Simu.) and each dot is an average of 50 randomly constructed communities with the same set of community parameters. Lines are results from theoretical approximation (Approx.). In **a** & **c**, $s = 5$. In **b** & **d**, $s = 0.5$. Other community parameters in (a) and (b) are $S = 200$, $C = 0.3$, $\sigma = 0.1$, $\sigma_x = 0.1$. Other community parameters in (c) and (d) are $S = 200$, $C = 0.3$, $\sigma = 0.1$, $\mu_x = 2$.

Supplemental Fig. S23. Simulated results on the relationship between the reactivity of A and the reactivity of M . **a-d**, Randomly generated communities with species abundances sampled from different random distributions. Blue dots in the grey region (i.e., Quadrant 1) indicate that both A and M are reactive. Dots in the red region (i.e., Quadrant 2) indicate that A is non-reactive while M is reactive. Dots in the green region (i.e., Quadrant 3) indicate that both A and M are non-reactive. Dots in the white region (i.e., Quadrant 4) indicate that A is reactive while M is non-reactive. **e**, Proportions of dots in different quadrants of (a)-(d). Details about the community generation strategy are provided in Supplementary Note 8. These simulation results suggest that when A is reactive, then the corresponding M is reactive. However, when A is non-reactive, the corresponding M can be reactive or non-reactive.

Minor point:

My last point concerns the effect of repeated perturbations. I think studying the correlation between measures of variability (see Arnoldi et al, Ecollet 2019) reactivity and asymptotic stability, would be more elegant than looking at simulation model with always the same perturbation. Indeed, those measures can all be derived from the matrix M (see Arnoldi et al, JTB I 2016). Also, it can be shown that variability integrates the whole transient response so it thus providing a mathematical explanation as to why the latter may depend more on reactivity than on the asymptotic response.

Thank you for this comment. Inspired by this and your work, we now construct a perturbed ecological community model and explore the correlation between variability, reactivity, and asymptotic stability. By conducting numerical simulations under different types of perturbations (i.e., immigration-type perturbation, demographic-type perturbation, and environmental-type perturbation), we find that compared with non-reactive communities, the variability of a stable

reactive community is relatively high (see Supplemental Figs. S15-S18 attached below). Since variability can be an indicator of the risk that an ecosystem experiencing species loss and system collapse, this result further supports our claim that with frequent perturbations, reactivity becomes central in predicting species' persistence.

We discuss this new work with a new paragraph in the Results section, a new Supplementary Note and four new Supplemental Figures.

The new paragraph in the Results is as follows.

L297-L308: *“Frequent perturbations can also be modeled through the theoretical framework for modelling ecological variability [53] of Arnoldi et al. This modelling framework is a linearised model and is thus valid for systems operating near equilibrium. As a result, it is not suited for studying extinctions directly, as these tend to occur when a system is far from equilibrium. Nevertheless, one can use variability in abundances over time to capture transient responses to perturbation, where the magnitude reflects the tendency of a community to change in time. In this way, variability can be a good indicator of the risk that an ecosystem will experience species loss and system collapse [28, 53]. Inspired by this approach [53], we constructed a perturbed community model and explored the relationship between variability and reactivity (Supplementary Note 7). This reveals that, as compared with non-reactive communities, the variability of a stable reactive community is high, which is consistent with our finding that stable reactive communities are more vulnerable to species loss and system collapse under frequent perturbations (Figs. S15-S18).”*

The new Supplemental Figures are as follows.

Supplemental Fig. S15. Responses of different perturbed linearised food web models under immigration type perturbations ($\alpha = 0$). Each line in this figure captures the change in abundance of each species. \mathcal{V} quantifies the degree of variability, while \mathcal{I} quantifies the degree of invariability. Compared with non-reactive stable communities (upper row), reactive stable communities have relatively high variability (correspondingly, relatively low invariability), indicating these communities are more prone to experience species loss or system collapse. Community parameters are: $S = 50$, $C = 0.2$, $\sigma = 0.05$. For non-reactive communities, we have $d = 1$. For reactive communities, we have $d = 0.1$. σ_p of each species is sampled from a uniform distribution $U[0.15,0.25]$.

Supplemental Fig. S16. Correlation between stability, reactivity and variability under immigration-type perturbations ($\alpha = 0$). Each dot in this figure is an average of 50 randomly constructed communities for the specific parameter combination. Each error bar denotes the standard deviation of these 50 communities. Community parameters are: $S = 50$, $C = 0.2$, $\sigma = 0.05$. σ_p of each species is sampled from a uniform distribution $U[0.15, 0.25]$.

Supplemental Fig. S17. Correlation between stability, reactivity and variability under demographic-type perturbations ($\alpha = 1$). Each dot in this figure is an average of 50 randomly constructed communities for the specific parameter combination. Each error bar denotes the standard deviation of these 50 communities. Community

parameters are: $S = 50$, $C = 0.2$, $\sigma = 0.05$. σ_p of each species is sampled from a uniform distribution $U[0.15,0.25]$.

Supplemental Fig. S18. Correlation between stability, reactivity and variability under environmental-type perturbations ($\alpha = 2$). Each dot in this figure is an average of 50 randomly constructed communities for the specific parameter combination. Each error bar denotes the standard deviation of these 50 communities. Community parameters are: $S = 50$, $C = 0.2$, $\sigma = 0.05$. σ_p of each species is sampled from a uniform distribution $U[0.15,0.25]$.

Thank you again for your careful review and helpful comments.

Response to Reviewer 3

The authors have made substantial revisions to the manuscript in response to the first round of refereeing, including adding quite a bit of new material to expand the results. My personal view is that the work as a still whole does not meet the threshold of novelty for publication in Nature Comms, however I understand that the other referees do not share this opinion and of course it is ultimately up to the editor to decide.

Thank you for your careful review. Below we provide point-by-point responses to each of your comments in order.

I need to highlight one major issue that must be addressed before publication can be considered. I believe the analysis presented in Figure 2c has serious problems. The figure is presented as a validation of the reactivity theory for empirical communities - this is a major selling point of the paper and this figure is the only place where this claim is quantified. The "analytical prediction" red dots sit perfectly on top of experimental data in a way that is highly suspicious to anyone who has worked with real data in this setting, so I have attempted to reproduce this figure. I have not been able to reproduce the figure - my calculations (using the formulae from the manuscript and the data from the references) are some way off what is reported in the figure, although numbers I get for the experimental data are close so I think I am doing the right thing. The authors must provide a complete and detailed procedure for producing the "analytical prediction" red dots, so that others can reproduce it. I think the current figure is incorrect and misrepresents the results, and the paper cannot be published until this issues is resolved.

The reason the agreement between the theory and data here is so shocking is that the empirical communities only have a small number of species in them. Reference 38 only reported a community matrix of size 8x8. But as everyone who understands the body of RMT theory that this paper users should know, the results are only valid in the limit of large numbers of communities. Indeed, figure 2b includes up to $S=1500$, but one has to delve into the supplement to discover that figure 2c has just a handful of species. The authors should know that the theory they use does not apply in this setting and they should not claim that it does.

These are good and important points. We discussed them at length and went back to the way that the 'red dot' analysis was done, and fully agree that its role was overstated in the text. This analysis was originally intended to simply show that the model can generate reasonable values that fit with the data. This was done by first fitting gamma distributions to the possible steady state abundances of each experimental system, then exploring the range of potential reactivity values predicted under these distributions to see whether there were values that could capture the "true" reactivity of each system within the 95% confidence intervals of the estimated parameters of the Gamma distribution. Having reviewed this process and our manuscript, we entirely agree that this was not presented clearly enough in our main text and, moreover, it is not a strong test of our theory. To avoid confusion, therefore, we have now cut this aspect of the data analysis.

Instead, we focus on using the data to qualitatively test our key broad scale predictions. Namely that a) reactivity is common in natural systems and b) it is associated with a mixture of interaction types. Importantly, in doing so we now also make clearer to the readers that the available data comes from communities that are relatively small, while RMT is designed for larger communities. Finally, we would like to re-emphasize that this exploration of experimental data is not intended to be the major selling point of our work - precisely because the data available are so limited! You will note, for example, that we do not discuss these results in the abstract. Nevertheless, we think it is nice to include it, given the patterns seen are consistent with our predictions.

The relevant revised Results paragraph now reads:

L199-L210: “Our work suggests that reactivity is common in nature. To evaluate this key prediction, we sought data from microbial communities that allow reactivity to be estimated. We identified seven communities, two found within the mammalian gut [42, 43], four isolated from the soil [44], and one arbitrarily and artificially assembled in vitro using a mixture of gut and soil associated microbes [45] (see Supplementary Note 4). We note that these communities are of low diversity (4 or 5 taxa) as compared to those studied with our theory, which is designed for diverse communities. Nevertheless, in line with our key prediction that reactivity is important, we find evidence of reactivity in all of the natural communities (Fig. 3d). Moreover, these natural communities also display a mixture of interaction types, which we again find is associated with stable reactive communities in the theory (Fig. 3d). By contrast, the one artificial community that contains only mutualistic and competitive interactions is non-reactive, which is again consistent with our prediction that these interaction types are less likely to lead to stable reactive systems (Fig. 3d).”

The revised paragraph in the Methods section is as follows.

L425-L432: “**Reactivity analysis of empirical microbial communities.** We analyse the reactivity of seven different microbial communities [42-45] from previous empirical studies. For these communities, the researchers have already inferred the true interaction networks (A_{ij}) and intrinsic growth rates (r_i). For a given community, we then perform the following steps: 1) Using the empirically derived A_{ij} and r_i parameters, we identify the underlying equilibrium abundances of each species within the community, then determine the empirical community matrix \mathbf{M}_e . 2) We then determine the reactivity directly from \mathbf{M}_e , via calculating the eigenvalues of $\mathbf{H}_e = (\mathbf{M}_e + \mathbf{M}_e^T)/2$ (blue bars, left part of Fig. 3d).”

The revised Figures are as follows (note, we have now moved the empirical analysis to Fig 3).

Fig. 2. Reactivity criteria for large complex ecosystems. **a**, Predicting the eigenvalue distribution of H and corresponding reactivity profiles. In the first row, the maximum (grey circles) and minimum eigenvalues (grey diamonds) of 50 randomly generated communities are plotted. Blue and red lines are theoretical predictions of maximum and minimum eigenvalues, respectively. In the second row, we systematically vary C to obtain $\sigma\sqrt{SC}$ spanning $[0,1.5]$. Orange dots represent the percentage of reactive communities out of 100 samples from numerical simulations. Blue lines are the corresponding theoretical predictions for the numerical percentage. Grey regions show unstable communities. In all cases, phase transitions from non-reactivity to reactivity are well predicted by our theory. **b**, Reactivity criteria for different types of communities. Curves with different colours are critical $C - S$ curves for different communities, and combinations of S and C below each curve lead to non-reactive communities. Different types of communities form a strict hierarchy from predator-prey communities (most likely to be non-reactive) to mutualistic communities (most likely to be reactive). **c**, Extension of our theory to communities with mixed interaction types. Left part of this panel gives theoretical predictions and right part of this panel shows results from numerical simulations. Each data point in the right part is an average of 50 randomly generated communities with the same set of parameters. In this figure, we have $d = 1$. In the first row of (a), we have $S = 150$, $C = 0.25$, $\sigma = 0.25$. In the second row of (a), other parameters are the same as the first row except for the varying C . In (b), we have $\sigma = 0.1$. In (c), we have $S = 100$, $C = 0.1$, $\sigma = 0.05$.

Fig. 3. Measuring the distance between reactivity and instability. **a**, Comparison of the critical $C - S$ curves of reactivity and stability for communities with different interaction types. The introduction of reactivity criteria (blue lines) and stability criteria (red lines) may divide the $C - S$ plane into 3 parts, where the blue region leads to stable and non-reactive communities, the red region leads to stable and reactive communities, and the grey region leads to unstable and reactive communities. **b**, A general view of the distance between reactivity and instability. Here we define the area ratio of the reactive stable region (red region in (a)) to the whole stable region (red and blue regions in (a)) as the normalized distance between reactivity and instability. Yellow bars are theoretical results, while green dots are numerical results. Each green dot is obtained by measuring the corresponding area ratio in a simulated phase transition diagram (i.e., simulated version of a). **c**, Distance between reactivity and instability for communities with mixed types of interaction (i.e., predator-prey interactions, mutualistic interactions, and competitive interactions). When the connectance (or community size) is fixed, the distance is measured by the critical ratio of the number of species (or the ratio of connectance) based on that in a. Here we plot this distance for three cases, two of these have fixed community size (left panel and middle panel) while the other one has fixed connectance (right panel). As the proportion of predator-prey interactions ($P_{+/-}$) increases, the distance increases. As the proportion of mutualistic interactions ($P_{+/+}$) or competitive interactions ($P_{-/-}$) increases, the distance decreases. And these are consistent with the results given in (b). **d**, Reactivity analysis of empirical microbial communities. Left part shows the reactivity calculated directly from experimental data, and right part shows

proportions of different types of interactions in these communities. Here communities 1 and 2 are mouse microbial communities [42, 43], communities 3 to 6 are soil microbial communities [44], and community 7 is a randomly constructed community [45]. In (a)-(c), we set $d = 1$, $\sigma = 0.2$.

Unfortunately this fits a pattern of overstating the strength of their results. For example, in the abstract it is claimed that they "develop a new body of theory to understand and predict how any large complex ecosystem will behave immediately after perturbations"; a correct statement would be that they "apply existing theory to predict how some large linear systems respond to small perturbations". I strongly recommend the authors to carefully check the manuscript for other similar exaggerations.

In the abstract we now say, "Using random matrix theory, we study how complex ecosystems behave immediately after small perturbations". And, in the last paragraph of the Introduction section, we now say "Based on recent development of random matrix theory [4, 6, 8, 11, 13, 37-39], here we develop theory to predict when reactivity is expected in complex communities and how this relates to their stability". We think these are reasonable and hope they sound ok to you too.

Thank you again for your helpful comments. The manuscript is certainly stronger as a result.

REVIEWERS' COMMENTS

Reviewer #2 (Remarks to the Author):

I have no further comments. I thank the authors for taking the time to carefully and convincingly answer my previous remarks.

Reviewer #3 (Remarks to the Author):

The authors have now removed the empirical analysis that I had serious concerns about previously, and they have added some caveats to the text about the number of taxa in the data they present. This change has further reduced the novelty of the work, but I think it has fixed the biggest problems of the previous draft. I still don't recommend publication, but equally I don't foresee any major problems around reproducibility if you do decide to publish.